# A Unified Bayesian Framework for Discriminative and Generative Continual Learning

## Abstract

Continual Learning is a learning paradigm where learning systems are trained on a sequence of tasks. The goal here is to perform well on the current task without suffering from a performance drop on the previous tasks. Two notable directions among the recent advances in continual learning with neural networks are (1) variational Bayes based regularization by learning priors from previous tasks, and, (2) learning the structure of deep networks to adapt to new tasks. So far, these two approaches have been orthogonal. We present a novel Bayesian framework for continual learning based on learning the structure of deep neural networks, addressing the shortcomings of both these approaches. The proposed framework learns the deep structure for each task by learning which weights to be used, and supports inter-task transfer through the overlapping of different sparse subsets of weights learned by different tasks. An appealing aspect of our proposed continual learning framework is that it is applicable to both discriminative (supervised) and generative (unsupervised) settings. Experimental results on supervised and unsupervised benchmarks shows that our model performs comparably or better than recent advances in continual learning.

## 1 Introduction

Continual learning (CL) (Ring, 1997; Parisi et al., 2019) is the learning paradigm where a single model is subjected to a sequence of tasks. At any point of time, the model is expected to ($i$) make predictions for the tasks it has seen so far, ($ii$) if subjected to training data for a new task, adapt to the new task leveraging the past knowledge if possible (forward transfer) and benefit the previous tasks if possible (backward transfer). While the desirable aspects of more mainstream transfer learning (sharing of bias between related tasks (Pan & Yang, 2009)) might reasonably be expected here too, the principal challenge is to retain the predictive power for the older tasks even after learning new tasks, thus avoiding the so-called *catastrophic forgetting*.

Real world applications in, for example, robotics or time-series forecasting, are rife with this challenging learning scenario, the ability to adapt to dynamically changing environments or evolving data distributions being essential in these domains. Continual learning is also desirable in unsupervised learning problems as well (Smith et al., 2019; Rao et al., 2019b) where the goal is to learn the underlying structure or latent representation of the data. Also, as a skill innate to humans (Flesch et al., 2018), it is naturally an interesting scientific problem to reproduce the same capability in artificial predictive modelling systems.

Existing approaches to continual learning are mainly based on three foundational ideas. One of them is to constrain the parameter values to not deviate significantly from their previously learned value by using some form of regularization or trade-off between previous and new learned weights (Schwarz et al., 2018; Kirkpatrick et al., 2017; Zenke et al., 2017; Lee et al., 2017). A natural way to accomplish this is to train a model using online Bayesian inference, whereby the posterior of the parameters learned from task $t$ serve as the prior for task $t + 1$ as in Nguyen et al. (2018) and Zeno et al. (2018). This new informed prior helps in the forward transfer, and also prevents catastrophic forgetting by penalizing large deviations from itself. In particular, VCL (Nguyen et al., 2018) achieves the state of the art results by applying this simple idea to Bayesian neural networks. The second idea is to perform an incremental model selection for every new task. For neural networks, this is done by evolving the structure as newer tasks are encountered (Golkar et al., 2019; Li

et al., 2019). Structural learning is a very sensible direction in continual learning as a new task may require a different network structure than old unrelated tasks and even if the tasks are highly related their lower layer representations can be very different. Another advantage of structural learning is that while retaining a shared set of parameters (which can be used to model task relationships) it also allow task-specific parameters that can increase the performance of the new task while avoiding catastrophic forgetting caused due to forced sharing of parameters. The third idea is to invoke a form of 'replay', whereby selected or generated samples representative of previous tasks, are used to retrain the model after new tasks are learned.

In this work, we introduce a novel Bayesian nonparametric approach to continual learning that seeks to incorporate the ability of structure learning into the simple yet effective framework of online Bayes. In particular, our approach models each hidden layer of the neural network using the Indian Buffet Process (Griffiths & Ghahramani, 2011) prior, which enables us to learn the network structure as new tasks arrive continually. We can leverage the fact that any particular task $t$ uses a sparse subset of the connections of a neural network $N_t$, and different related tasks share different subsets (albeit possibly overlapping). Thus, in the setting of continual learning, it would be more effective if the network could accommodate changes in its connections dynamically to adapt to a newly arriving task. Moreover, in our model, we perform the automatic model selection where each task can select the number of nodes in each hidden layer. All this is done under the principled framework of variational Bayes and a nonparametric Bayesian modeling paradigm.

Another appealing aspect of our approach is that in contrast to some of the recent state-of-the-art continual learning models (Yoon et al., 2018; Li et al., 2019) that are specific to supervised learning problems, our approach applies to both deep discriminative networks (supervised learning) where each task can be modeled by a Bayesian neural network (Neal, 2012; Blundell et al., 2015), as well as deep generative networks (unsupervised learning) where each task can be modeled by a variational autoencoder (VAE) (Kingma & Welling, 2013).

## 2 PRELIMINARIES

**Bayesian neural networks** (Neal, 2012) are discriminative models where the goal is to model the relationship between inputs and outputs via a deep neural network with parameters $\boldsymbol{w}$. The network parameters are assumed to have a prior $p(\boldsymbol{w})$ and the goal is to infer the posterior given the observed data $\mathcal{D}$. The exact posterior inference is intractable in such models. One such approximate inference scheme is Bayes-by-Backprop (Blundell et al., 2015) that uses a mean-field variational posterior $q(\boldsymbol{w})$ over the weights. Reparameterized samples from this posterior are then used to approximate the lower bound via Monte Carlo sampling. Our goal in the continual learning setting is to learn such Bayesian neural networks for a sequence of tasks by inferring the posterior $q_t(\boldsymbol{w})$ for each task $t$, without forgetting the information contained in the posteriors of previous tasks.

**Variational autoencoders** (Kingma & Welling, 2013) are generative models where the goal is to model a set of inputs $\{\boldsymbol{x}\}_{n=1}^{N}$ in terms of a stochastic latent variables $\{\boldsymbol{z}\}_{n=1}^{N}$. The mapping from each $\boldsymbol{z}_n$ to $\boldsymbol{x}_n$ is defined by a generator/decoder model (modeled by a deep neural network with parameters $\theta$) and the reverse mapping is defined by a recognition/encoder model (modeled by another deep neural network with parameters $\phi$). Inference in VAEs is done by maximizing the variational lower bound on the marginal likelihood. It is customary to do point estimation for decoder parameters $\theta$ and posterior inference for encoder parameters $\phi$. However, in the continual learning setting, it would be more desirable to infer the full posterior $q_t(\boldsymbol{w})$ for each task's encoder and decoder parameters $\boldsymbol{w} = \{\theta, \phi\}$, while not forgetting the information about the previous tasks as more and more tasks are observed. Our proposed continual learning framework address this aspect as well.

**Variational Continual Learning** (VCL) Nguyen et al. (2018) is a recently proposed approach to continual learning that combats catastrophic forgetting in neural networks by modeling the network parameters $\boldsymbol{w}$ in a Bayesian fashion and by setting $p_t(\boldsymbol{w}) = q_{t-1}(\boldsymbol{w})$, that is, a task reuses the previous task's posterior as its prior. VCL solves the follow KL divergence minimization problem

$$q_t(\boldsymbol{w}) = \arg\min_{q \in \mathcal{Q}} \text{KL}\left(q(\boldsymbol{w}) \| \frac{1}{Z_t} q_{t-1}(\boldsymbol{w}) p(\mathcal{D}_t | \boldsymbol{w})\right) \tag{1}$$

While offering a principled way that is applicable to both supervised (discriminative) and unsupervised (generative) learning settings, VCL assumes that the model structure is held fixed throughout,

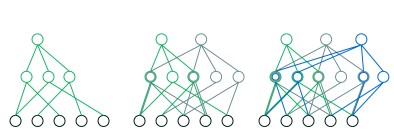 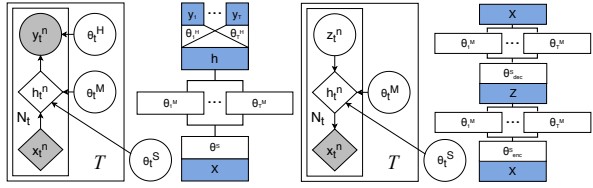

(a) Illustration on single hidden layer          (b) (i)Discriminative Model    (ii) Generative Model (VAE)

Figure 1: (**a**) Evolution of network structure for 3 consecutive tasks. Weights used by a task are denoted with respective colors. Note that there can be overlapping of structure between tasks. (**b**) Schematics representing our models, $\theta_S$ are parameters shared across all task, $\theta^M$ are task specific mask parameters, and $\theta^H$ are last layer separate head parameters. In our exposition, we collectively denote these parameters by $\boldsymbol{W} = \boldsymbol{B} \odot \boldsymbol{V}$ with the masks being $\boldsymbol{B}$ and other parameters being $\boldsymbol{V}$.

which can be limiting in continual learning where the number of tasks and their complexity is usually unknown beforehand. This necessitates adaptively inferring the model structure, that can potentially adapt with each incoming task. Another limitation of VCL is that the unsupervised version, based on performing CL on VAEs, only does so for the decoder model's parameters (shared by all tasks). It uses completely task-specific encoders and, consequently, is unable to transfer information across tasks in the encoder model. Our approach addresses both these limitations in a principled manner.

## 3  BAYESIAN STRUCTURE ADAPTATION FOR CONTINUAL LEARNING

In this section, we present a Bayesian model for continual learning that can potentially grow and adapt its structure as more and more tasks arrive. Our model extends seamlessly for unsupervised learning as well. For brevity of exposition, in this section, we mainly focus on the supervised setting where a task has labeled data with known task identities $t$ (task-incremental). We then briefly discuss the unsupervised extension (based on VAEs) in Sec. 3.3 where task boundaries may or may not (task-agnostic) be available and provide further details in the appendix (Sec I).

Our approach uses a basic primitive that models each hidden layer using a nonparametric Bayesian prior (Fig. 1a shows an illustration and Fig. 1b shows a schematic diagram). We can use these hidden layers to model feedforward connections in Bayesian neural networks or VAE models. For simplicity, we will assume a single hidden layer, the first task activates as many hidden nodes as required and learns the posterior over the subset of edge weights incident on each active node. Each subsequent task reuses some of the edges learned by the previous task and uses the posterior over the weights learned by the previous task as the prior. Additionally, it may activate some new nodes and learn the posterior over some of their incident edges. It thus learns the posterior over a subset of weights that may overlap with weights learned by previous tasks. While making predictions, a task uses only the connections it has learned. More slack for later tasks in terms of model size (allowing it to create new nodes) indirectly lets the task learn better without deviating too much from the prior (in this case, posterior of the previous tasks) and further reduces chances of *catastrophic forgetting* (Kirkpatrick et al., 2017).

### 3.1  GENERATIVE STORY.

Omitting the task id $t$ for brevity, consider modeling $t^{th}$ task using a neural network having $L$ hidden layers. We model the weights in layer $l$ as $\boldsymbol{W}^l = \boldsymbol{B}^l \odot \boldsymbol{V}^l$, a point-wise multiplication of a real-valued matrix $\boldsymbol{V}^l$ (with a Gaussian prior $\mathcal{N}(0, \sigma_0^2)$ on each entry) and a task-specific binary matrix $\boldsymbol{B}^l$. This ensures sparse connection weights between the layers. Moreover, we model $\boldsymbol{B}^l \sim \mathrm{IBP}(\alpha)$ using the Indian Buffet Process (IBP) Griffiths & Ghahramani (2011) prior, where the hyperparameter $\alpha$ controls the number of nonzero columns in $B$ and its sparsity. The IBP prior thus enables learning the size of $\boldsymbol{B}^l$ (and consequently of $\boldsymbol{V}^l$) from data. As a result, the number of nodes in the hidden layer is learned adaptively from data. The output layer weights are denoted as $\boldsymbol{W}_{out}$ with each weight having a Gaussian prior $\mathcal{N}(0, \sigma_0^2)$. The outputs are

$$\boldsymbol{y}_n \sim \mathrm{Lik}(\boldsymbol{W}_{out}\boldsymbol{\phi}_{NN}(\boldsymbol{x}_n)), n = 1, \dots, N \tag{2}$$

Here $\boldsymbol{\phi}_{NN}$ is the function computed (using parameter samples) up to the last hidden layer of the network thus formed, and $\mathrm{Lik}$ denotes the likelihood model for the outputs.

Similar priors on the network weights have been used in other recent works to learn sparse deep neural networks (Panousis et al., 2019; Xu et al., 2019). However, these works assume a single task to be learned. In contrast, our focus here is to leverage such priors in the continual learning setting where we need to learn a sequence of tasks while avoiding the problem of catastrophic forgetting. Henceforth, we further suppress the superscript denoting layer number from the notation for simplicity; the discussion will hold identically for all hidden layers. When adapting to a new task, the posterior of $V$ learned from previous tasks is used as the prior. A new $B$ is learned afresh, to ensure that a task only learns the subset of weights relevant to it.

**Stick Breaking Construction.** As described before, to adaptively infer the number of nodes in each hidden layer, we use the IBP prior (Griffiths & Ghahramani, 2011), whose truncated stick-breaking process (Doshi et al., 2009) construction for each entry of $B$ is as follows

$$\nu_k \sim \text{Beta}(\alpha, 1), \quad \pi_k = \prod_{i=1}^{k} \nu_i, \quad B_{d,k} \sim \text{Bernoulli}(\pi_k) \tag{3}$$

for $d \in 1, ..., D$, where $D$ denotes the number of input nodes for this hidden layer, and $k \in 1, 2, ..., K$, where $K$ is the truncation level and $\alpha$ controls the effective value of $K$, i.e., the number of active hidden nodes. Note that the prior probability $\pi_k$ of weights incident on hidden node $k$ being nonzero decreases monotonically with $k$, until, say, $K$ nodes, after which no further nodes have any incoming edges with nonzero weights from the previous layer, which amounts to them being turned off from the structure. Moreover, due to the cumulative product based construction of the $\pi_k$'s, an implicit ordering is imposed on the nodes being used. This ordering is preserved across tasks, and allocation of nodes to a task follows this, facilitating reuse of weights.

The truncated stick-breaking approximation is a practically plausible and intuitive solution for continual learning since a fundamental tenet of continual learning is that the model complexity should not increase in an unbounded manner as more tasks are encountered. Suppose we fix a budget on the maximum allowed size of the network (no. hidden nodes in a layer) after it has seen, say, $T$ tasks. Which exactly corresponds to the truncation level for each layer. Then for each task, nodes are allocated conservatively from this total budget, in a fixed order, conveniently controlled by the $\alpha$ hyperparameter. In appendix (Sec. D), we also discuss a dynamic expansion scheme that avoids specifying a truncation level (and provide experimental results).

## 3.2 INFERENCE

Exact inference is intractable in this model due to non-conjugacy. Therefore, we resort to the variational inference (Blei et al., 2017). We employ structured mean-field approximation (Hoffman & Blei, 2015), which performs better than normally used mean-field approximation, as the former captures the dependencies in the approximate posterior distributions of $B$ and $\nu$. In particular, we use $q(V, B, v) = q(V)q(B|v)q(v)$, where, $q(V) = \prod_{d=1}^{D} \prod_{k=1}^{K} \mathcal{N}(V_{d,k}|\mu_{d,k}, \sigma_{d,k}^2)$ is mean field Gaussian approximation for network weights. Corresponding to the Beta-Bernoulli hierarchy of (3), we use the conditionally factorized variational posterior family, that is, $q(B|v) = \prod_{d=1}^{D} \prod_{k=1}^{K} \text{Bern}(B_{d,k}|\theta_{d,k})$, where $\theta_{d,k} = \sigma(\rho_{d,k} + logit(\pi_k))$ and $q(v) = \prod_{k=1}^{K} \text{Beta}(v_k|\nu_{k,1}, \nu_{k,2})$. Thus we have $\Theta = \{\nu_{k,1}, \nu_{k,2}, \{\mu_{d,k}, \sigma_{d,k}, \rho_{d,k}\}_{d=1}^{D}\}_{k=1}^{K}$ as set of learnable variational parameters.

Each column of $B$ represents the binary mask for the weights incident to a particular node. Note that although these binary variables (in a single column of $B$) share a common prior, the posterior for each of these variables are different, thereby allowing a task to selectively choose a subset of the weights, with the common prior controlling the degree of sparsity.

$$\mathcal{L} = \mathbb{E}_{q(V,B,v)}[\ln p(Y|V, B, v)] - KL(q(V, B, v)||p(V, B, v)) \tag{4}$$

$$\mathcal{L} = \frac{1}{S} \sum_{i=1}^{S} [f(V^i, B^i, v^i) - KL[q(B|v^i)||p(B|v^i)]] - KL[q(V)||p(V)] - KL[q(v)||p(v)] \tag{5}$$

Bayes-by-backprop (Blundell et al., 2015) is a common choice for performing variational inference in this context. Eq. 4 defines the Evidence Lower Bound (ELBO) in terms of data-dependent likelihood and data-independent KL terms which further gets decomposed using mean-field factorization.

The expectation terms are optimized by unbiased gradients from the respective posteriors. All the KL divergence terms in (Eq. 5) have closed form expressions; hence using them directly rather than estimating them from Monte Carlo samples alleviates the approximation error as well as the computational overhead, to some extent. The log-likelihood term can be decomposed as

$$f(\boldsymbol{V}, \boldsymbol{B}, \boldsymbol{v}) = \log \mathrm{Lik}(\boldsymbol{Y}|\boldsymbol{V}, \boldsymbol{B}, \boldsymbol{v}) = \log \mathrm{Lik}(\boldsymbol{Y}|\boldsymbol{W}_{out}\phi_{NN}(\boldsymbol{X}; \boldsymbol{V}, \boldsymbol{B}, \boldsymbol{v})) \qquad (6)$$

where $(\boldsymbol{X}, \boldsymbol{Y})$ is the training data. For regression, Lik can be Gaussian with some noise variance, while for classification it can be Bernoulli with a probit or logistic link. Details of sampling gradient computation for terms involving beta and Bernoulli r.v.'s is provided in the appendix. (Sec. F).

### 3.3 Unsupervised Continual Learning

Our discussion thus far has primarily focused on continual learning where each task is a supervised learning problem. Our framework however readily extends to unsupervised continual learning (Nguyen et al., 2018; Smith et al., 2019; Rao et al., 2019b) where we assume that each task involves learning a deep generative model, commonly a VAE. In this case, each input observation $\boldsymbol{x}_n$ has an associated latent variable $\boldsymbol{z}_n$. Collectively denoting all inputs as $\boldsymbol{X}$ and all latent variables as $\boldsymbol{Z}$, we can define ELBO similar to Eq. 4 as

$$\mathcal{L} = \mathbb{E}_{q(\boldsymbol{Z}, \boldsymbol{V}, \boldsymbol{B}, \boldsymbol{v})}[\ln p(\boldsymbol{X}|\boldsymbol{Z}, \boldsymbol{V}, \boldsymbol{B}, \boldsymbol{v})] - KL(q(\boldsymbol{Z}, \boldsymbol{V}, \boldsymbol{B}, \boldsymbol{v})||p(\boldsymbol{Z}, \boldsymbol{V}, \boldsymbol{B}, \boldsymbol{v})) \qquad (7)$$

Note that, unlike the supervised case, the above ELBO also involves an expectation over $\boldsymbol{Z}$. Similar to Eq. 5 this can be approximated using Monte Carlo samples, where each $\boldsymbol{z}_n$ is sampled from the amortized posterior $q(\boldsymbol{z}_n|\boldsymbol{V}, \boldsymbol{B}, \boldsymbol{v}, \boldsymbol{x}_n)$. In addition to learning the model size adaptively, as shown in the schematic diagram (Fig. 1b (ii)), our model learns shared weights and task-specific masks for the encoder and decoder models. In contrast, VCL uses fixed-sized model with entirely task-specific encoders, which prevents knowledge transfer across the different encoders.

### 3.4 Other Key Considerations

**Task Agnostic Setting** Our framework extends to task-agnostic continual learning as well where the task boundaries are unknown. Based on Lee et al. (2020), we use a gating mechanism (Eq. 8 with $t_n$ represents the task identity of $n^{th}$ sample $x_n$) and define marginal log likelihood as

$$p(\boldsymbol{t}_n = k|\boldsymbol{x}_n) = p(\boldsymbol{x}_n|\boldsymbol{t}_n = k)p(\boldsymbol{t}_n = k) \bigg/ \sum_{k=1}^{K} p(\boldsymbol{x}_n|\boldsymbol{t}_n = k)p(\boldsymbol{t}_n = k) \qquad (8)$$

$$\log p(\boldsymbol{X}) = \mathbb{E}_{q(t=k)}\left[p(\boldsymbol{X}, \boldsymbol{t} = k|\theta)\right] + KL\left(q(\boldsymbol{t} = k)||p(\boldsymbol{t} = k|\boldsymbol{X}, \theta)\right) \qquad (9)$$

where, $q(t = k)$ is the variational posterior over task identity. Similar to E-step in Expectation Maximization (Moon, 1996), we can reduce the KL-Divergence term to zero and get the M-step as

$$\arg\max_{\theta} \log p(\boldsymbol{X}) = \arg\max_{\theta} \mathbb{E}_{p(\boldsymbol{t}=k|\boldsymbol{X}, \theta_{\mathrm{old}})} \log p(\boldsymbol{X}|\boldsymbol{t} = k) \qquad (10)$$

Here, $\log p(\boldsymbol{X}|\boldsymbol{t} = k)$ is intractable but can be replaced with its variational lower bound (Eq. 7). We use Monte Carlo sampling for approximating $p(\boldsymbol{x}_n|\boldsymbol{t}_n = k)$. Detecting samples from a new task is done using a threshold (Rao et al., 2019a) on the evidence lower bound (Appendix Sec. J)

**Masked Priors** Using the previous task's posterior as the prior for current task (Nguyen et al., 2018) may introduce undesired regularization in case of partially trained parameters that do not contribute to previous tasks and may promote catastrophic forgetting. Also, the choice of the initial prior as Gaussian leads to creation of more nodes than required due to regularization. To address this, we mask the new prior for the next task $t$ with the initial prior $p_t$ defined as

$$p_t(\boldsymbol{V}_{d,k}) = B_{d,k}^o q_{t-1}(\boldsymbol{V}_{d,k}) + (1 - B_{d,k}^o)p_0(\boldsymbol{V}_{d,k}) \qquad (11)$$

where $\boldsymbol{B}^o$ is the overall combined mask from all previously learned tasks i.e., $(\boldsymbol{B}^1 \cup \boldsymbol{B}^2 ... \cup \boldsymbol{B}^{t-1})$, $q_{t-1}, p_t$ are the previous posterior and current prior, respectively, and $p_0$ is the prior used for the first task. The standard choice of initial prior $p_0$ can be a uniform distribution.

# 4 RELATED WORK

One of the key challenges in continual learning is to prevent catastrophic forgetting, typically addressed through regularization of the parameter updates, preventing them from drastically changing from the value learnt from the previous task(s). Notable methods based on this strategy include EwC (Kirkpatrick et al., 2017), SI (Zenke et al., 2017), LP (Smola et al., 2003), etc. Superseding these methods is the Bayesian approach, a natural remedy of catastrophic forgetting in that, for any task, the posterior of the model learnt from the previous task serves as the prior for the current task, which is the canonical online Bayes. This approach is used in recent works like VCL (Nguyen et al., 2018) and task agnostic variational Bayes (Zeno et al., 2018) for learning Bayesian neural networks in the CL setting. Our work is most similar in spirit to and builds upon this body of work.

Another key aspect in CL methods is *replay*, where some samples from previous tasks are used to fine-tune the model after learning a new task (thus refreshing its memory in some sense and avoiding catastrophic forgetting). Some of the works using this idea include Lopez-Paz et al. (2017), which solves a constrained optimization problem at each task, the constraint being that the loss should decrease monotonically on a heuristically selected replay buffer; Hu et al. (2019), which uses a partially shared parameter space for inter-task transfer and *generates* the replay samples through a data-generative module; and Titsias et al. (2020), which learns a Gaussian process for each task, with a shared mean function in the form a feedforward neural network, the replay buffer being the set of inducing points typically used to speed up GP inference. For VCL and our work, the coreset serves as a replay buffer (Appx. C); but we emphasize that it is not the primary mechanism to overcome catastrophic forgetting in these cases, but rather an additional mechanism to preventing it.

Recent work in CL has investigated allowing the structure of the model to dynamically change with newly arriving tasks. Among these, strong evidence in support of our assumptions can be found in Golkar et al. (2019), which also learns different sparse subsets of the weights of each layer of the network for different tasks. The sparsity is enforced by a combination of weighted $L_1$ regularization and threshold-based pruning. There are also methods that do not learn subset of weights but rather learn the subset of hidden layer nodes to be used for each task; such a strategy is adopted by either using Evolutionary Algorithms to select the node subsets (Fernando et al., 2017) or by training the network with task embedding based attention masks (Serrà et al., 2018). One recent approach Adel et al. (2020), instead of using binary masks, tries to adapt network weights at different scales for different tasks; it is also designed only for discriminative tasks.

Among other related work, Li et al. (2019); Yoon et al. (2018); Xu & Zhu (2018) either reuse the parameters of a layer, dynamically grows the size of the hidden layer, *or* spawn a new set of parameters (the model complexity being bounded through regularization terms or reward based reinforcements). Most of these approaches however tend to be rather expensive and rely on techniques, such as neural architecture search. In another recent work (simultaneous development with our work), Kessler et al. (2020) did a preliminary investigation on using IBP for continual learning. They however use IBP on hidden layer activations instead of weights (which they mention is worth considering), do not consider issues such as the ones we discussed in Sec. 3.4, and only applies to supervised setting. Modelling number active nodes for a given task has also been explored by Serrà et al. (2018); Fernando et al. (2017); Ahn et al. (2019), but modelling posterior over connections weights between these nodes achieves more sparsity and flexibility in terms of structural learning at the cost of increased number of parameters, von Oswald et al. (2020) tries to amortize the network parameters directly from input samples which is a promising direction and can be adapted for future research.

For non-stationary data, online variational Bayes is not directly applicable as it assumes independently and identically distributed (i.i.d.) data. As a result of which the variance in Gaussian posterior approximation will shrink with an increase in the size of training data, Kurle et al. (2020) proposed use of Bayesian forgetting, which can be naturally applied to our approach enabling it to work with non-stationary data but it requires some modifications for task-agnostic setup. In this work, we have not explored this extension keeping it as future work.

# 5 EXPERIMENTS

We perform experiments on both supervised and unsupervised CL and compare our method with relevant state-of-the-art methods. In addition to the quantitative (accuracy/log-likelihood compar-

isons) and qualitative (generation) results, we also examine the network structures learned by our model. Some of the details (e.g., experimental settings) have been moved to the appendix [1].

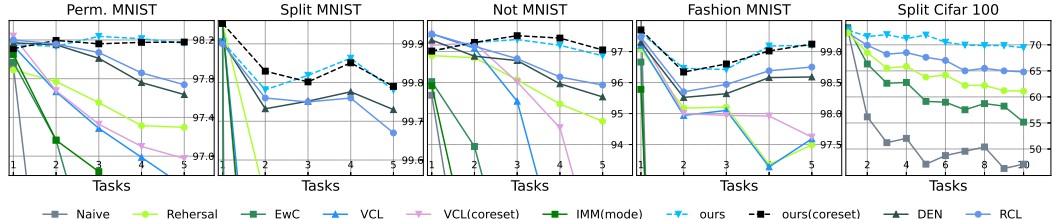

Figure 2: Mean test accuracies of tasks seen so far as newer tasks are observed on multiple benchmarks

## 5.1 SUPERVISED CONTINUAL LEARNING

We first evaluate our model on standard supervised CL benchmarks. We experiment with different existing approaches such as, Pure Rehearsal (Robins, 1995), EwC (Kirkpatrick et al., 2017), IMM (Lee et al., 2017), DEN (Yoon et al., 2018), RCL (Xu & Zhu, 2018), and "Naïve" which learns a shared model for all the tasks. We perform our evaluations on five supervised CL benchmarks: SplitMNIST, Split notMNIST(small), Permuted MNIST, Split fashionMNIST and Split Cifar100. The last layer heads (Appx. E.1) were kept separate for each task for fair baseline comparison.

For Split MNIST, Split notMNIST and Split fashionMNIST each dataset is split into 5 binary classification tasks. For Split Cifar100 the dataset was split 10 multiclass classification tasks. For Permuted MNIST, each task is a multiclass classification problem with a fixed random permutation applied to the pixels of every image. We generated 5 such tasks for our experiments.

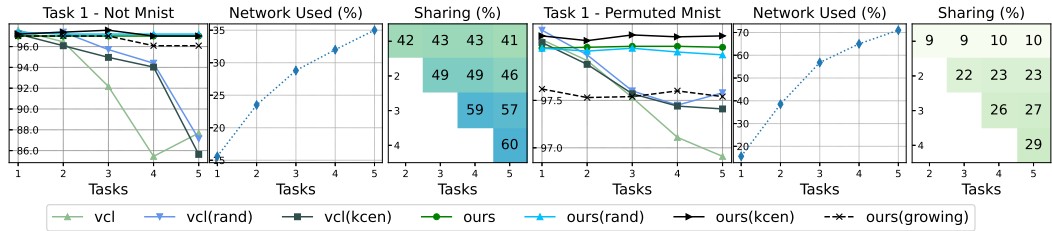

Figure 3: Test accuracy variation for first tasks (left), network used (middle) as new tasks are observed and percentage of network connections shared (right) among different tasks in first hidden layer

**Performance evaluation** Suppose we have a sequence of $T$ tasks. To gauge the effectiveness of our model towards preventing catastrophic forgetting, we report $(i)$ the test accuracy of first task after learning each of the subsequent tasks; and $(ii)$ the average test accuracy over all previous tasks $1, 2, \ldots t$ after learning each task $t$. For fair comparison, we use the same architecture for each of the baselines (details in Appx.), except for DEN and RCL that grows the structure size. We also report results on some additional CL metrics (Díaz-Rodríguez et al., 2018) in the Appx. (Sec. H.4).

Fig. 2 shows the mean test accuracies on all supervised benchmarks as new tasks are observed. As shown, the average test accuracy of our method (without as well as with coresets) is better than the compared baseline (here, we have used random point selection method for coresets). Moreover, the accuracy drops much more slowly than and other baselines showing the efficacy of our model in preventing catastrophic forgetting due to the adaptively learned structure. In Fig. 3, we show the accuracy on first task as new tasks arrive and compare specifically with VCL. In this case too, we observe that our method yields relatively stable first task accuracies as compared to VCL. We note that for permuted MNIST the accuracy of first task increases with training of new tasks which shows the presence of backward transfer, which is another desideratum of CL. We also report the performance with our dynamically growing network variant (for more details refer Appx. Sec. D).

---

[1]The code for our model can be found at this link: `https://github.com/npbcl/icml20`

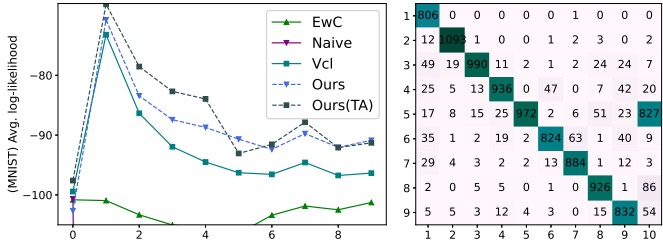

Figure 4: Sequential generation for MNIST (**left**) and notMNIST (**right**) datasets. Here $i^{th}$ row is generated after observing $i^{th}$ task (Appendix contains more illustrations and zoomed-in versions)

**Structural Observations** An appealing aspect of our work is that, the results reported above, which are competitive with the state-of-the-art, are achieved with very sparse neural network structures learnt by the model, which we analyze qualitatively here (Appendix Sec. H.1 shows some examples of network structures learnt by our model).

As shown in Fig. 3 (Network Used) IBP prior concentrates weights on very few nodes, and learns sparse structures. Also most newer tasks tend to allocate fewer weights and yet perform well, implying effective forward transfer. Another important observation as shown in Fig. 3 is that the weight sharing between similar tasks like notMNIST is a higher than that of non-similar tasks like permuted MNIST. Note that new tasks show higher weight sharing irrespective of similarity, this is an artifact induced by IBP (Sec 3.1) which tends to allocate more active weights on upper side of matrix.

We therefore conclude that although a new task tend to share weights learnt by old tasks, the new connections that it creates are indispensable for its performance. Intuitively, the more unrelated a task is to previously seen ones, the more new connections it will make, thus reducing *negative transfer* (an unrelated task adversely affecting other tasks) between tasks.

## 5.2 Unsupervised Continual Learning

We next evaluate our model on generative tasks under CL setting. For that, we compare our model with existing approaches such as Naïve, EwC and VCL. We do not include other methods mentioned in supervised setup as their implementation does not incorporate generative modeling. We perform continual learning experiments for deep generative models using a VAE style network. We consider two datasets, MNIST and notMNIST. For MNIST, the tasks are sequence of single digit generation from 0 to 9. Similarly, for notMNIST each task is one character generation from A to J. Note that,

Figure 5: Avg. log-likelihoods (left) for sequential generation (MNIST), confusion matrix (right) representing test samples mapped to each generative task learned in TA (task-agnostic) setting

| Methods | 3-NN | 5-NN |
|---|---|---|
| Naive | 30.1% | 33.1% |
| EwC | 16.6% | 19.5% |
| VCL | 16.0% | 19.1% |
| Ours | **0.37%** | **0.40%** |
| Ours (TA) | 5.79% | 5.32% |
| CURL (TA) | **4.58%** | **4.35%** |

Table 1: MNIST K-NN test error rates obtained in latent space for both task-agnostic and know task setting.

unlike VCL and other baselines where all tasks have separate encoder and a shared decoder, as we discuss in Sec. 3.3, our model uses a shared encoder for all tasks, but with task-specific masks for each encoder (cf., Fig. 1b (ii)). This enables transfer of knowledge while the task-specific mask effectively prevent catastrophic forgetting.

**Generation:** As shown in Fig 5, the modeling innovation we introduce for the unsupervised setting, results in much improved log-likelihood on held-out sets. In each individual figure in Fig 4, each row represents generated samples from all previously seen tasks and the current task. We see that the quality of generated samples in does not deteriorate as compared to other baselines as more tasks are encountered. This shows that our model can efficiently perform generative modeling by reusing subset of networks and creating minimal number of nodes for each task.

**Task-Agnostic Learning:** Fig 5 shows a particular case where nine tasks were inferred out of 10 class with high correlation among class 4 and 9 due to visual similarity between them. Since each task uses a set of network connection, this result enforces our models ability to model task relations based on network sharing. Further the log-likelihood obtained for task-agnostic setting is comparable to our model with known task boundaries, suggesting that our approach can be used effectively in task-agnostic settings as well.

**Representation Learning:** Table 1 represents the quality of the *unsupervisedly* learned representation by our unsupervised continual learning approach. For this experiment, we use the learned representations to train a KNN classification model with different K values. We note that despite having task-specific encoders VCL and other baselines fail to learn good latent representation, while the proposed model learns good representations when task boundaries are known and is comparable to state-of-the-art baseline CURL (Rao et al., 2019a) under task-agnostic setting.

## 6 CONCLUSION

We have successfully unified structure learning in neural networks with their variational inference in the setting of continual learning, demonstrating competitive performance with state-of-the-art models on both discriminative (supervised) and generative (unsupervised) learning problems. In this work, we have experimented with task-incremental continual learning for supervised setup and sequential generation task for unsupervised setting. we believe that our task-agnostic setup can be extended to class-incremental learning scenario where sample points from a set of classes arrives sequentially and model is expected to perform classification over all observed classes. It would also be interesting to generalize this idea to more sophisticated network architectures such as recurrent or residual neural networks, possibly by also exploring improved approximate inference methods. Few more interesting extensions would be in semi-supervised continual learning and continual learning with non-stationary data. Adapting other sparse Bayesian structure learning methods, e.g. Ghosh et al. (2018) to the continual learning setting is also a promising avenue. Adapting the *depth* of the network is a more challenging endeavour that might also be undertaken. We leave these extensions for future work.

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

## A    DATA

The data sets used in our experiments with train test split information are listed in table given below. MNIST dataset comprises $28 \times 28$ monochromatic images consisting of handwritten digits from 0 to 9. notMNIST dataset comprises of glyph's of letters A to J in different fonts formats with similar configuration as MNIST. fashion MNIST is also monochromatic comprising of 10 classes (T-shirt, Trouser, Pullover, Dress, Coat, Sandal, Shirt, Sneaker, Bag, Ankle boot) with similar to MNIST. Cifar100 dataset contains RGB images with 600 images per class.

| Dataset | Classes | Training size | Test size |
|---|---|---|---|
| MNIST | 10 | 60000 | 10000 |
| notMNIST | 10 | 14974 | 3750 |
| fashionMNIST | 10 | 50000 | 20000 |
| Cifar100 | 100 | 50000 | 10000 |

## B    MODEL CONFIGURATIONS

For permuted MNIST, split MNIST, split notMNIST and fashion MNIST experiments, we use fixed architecture of network for all the models with single hidden layer of 200 units except for DEN (which grows structure dynamically) which used two hidden layers initialized to $256, 128$ units.

The VCL implementation was taken from its official repository at `https://github.com/nvcuong/variational-continual-learning`. For DEN we used the official implementation `https://github.com/jaehong-yoon93/DEN`. IMM implementation was taken from `https://github.com/btjhjeon/IMM_tensorflow`, RCL implementation was taken from `https://https://github.com/xujinfan/Reinforced-Continual-Learning`, For EwC we used HAT's official implementation at `https://github.com/joansj/hat`. For others, we used our own implementations.

### B.1    SUPERVISED CONTINUAL LEARNING: HYPERPARAMETER SETTINGS

For MNIST, notMNIST, fashionMNIST datasets, our model uses single hidden layer neural network with 200 hidden units. For RCL (Xu & Zhu, 2018) and DEN (Yoon et al., 2018), two hidden layers were used with initial network size of 256, 128 units, respectively. For the Cifar100 dataset we used an Alex-net like structure with three convolutional layers of $128, 256, 512$ channels with $4 \times 4, 3 \times 3, 2 \times 2$ channels followed by two dense layers of $2048, 2048$ units each. For the convolutional layer, batch-norm layers were separate for each task. We adopt Adam optimizer for our model keeping a learning rate of 0.01 for the IBP posterior parameters and 0.001 for others; this is to avoid vanishing gradient problem introduced by sigmoid function. For selective finetuning, we use a learning rate of 0.0001 for all the parameters. The temperature hyperparameter of the Gumbel-softmax reparameterization for Bernoulli gets annealed from 10.0 to a minimum limit of 0.25. The value of $\alpha$ is initialized to 30.0 for the initial task and maximum of the obtained posterior shape parameters for each of subsequent tasks. Similar to VCL, we initialize our models with maximum-likelihood training for the first task. For all datasets, we train our model for 5 epochs. We selectively finetune our model after that for 5 epochs. For experiments including coresets, we use a coreset size of 50. Coreset selection is done using random and $k$-center methods Nguyen et al. (2018). For our model with dynamic expansion, we initialize our network with 50 hidden units.

### B.2    UNSUPERVISED CONTINUAL LEARNING: HYPERPARAMETER SETTINGS

For all datasets, our model uses 2 hidden layers with $500, 500$ units for encoder and symmetrically opposite for the decoder with a latent dimension of size 100 units. For other approaches like Naive, EwC and VCL (Kirkpatrick et al., 2017; Nguyen et al., 2018), we use task-specific encoders with 3 hidden layers of $500, 500, 500$ units respectively with latent size of 100 units, and a symmetrically reversed decoder with last two layers of decoder being shared among all the tasks and the first layer

being specific to each task. we use Adam optimizer for our model keeping the learning rate configuration similar to that of supervised setting. Temperature for gumbel-softmax reparametrization gets annealed from 10 to 0.25. We initialize encoder hidden layers $\alpha$ values as $40, 40$, respectively, and symmetrically opposite in decoder for the first task. We update $\alpha$'s in similar fashion to supervised setting for subsequent tasks. For latent layers, we intialize $\alpha$ to 20. For the unsupervised learning experiments, we did not use coresets.

## C  CORESET METHOD EXPLANATION

Proposed in Nguyen et al. (2018) as a method for cleverly sidestepping the issue of catastrophic forgetting, the coreset comprises representative training data samples from all tasks. Let $M^{(t-1)}$ denote the posterior state of the model before learning task $t$. With the $t$-th task's arrival having data $D_t$, a coreset $C_t$ is created comprising choicest examples from tasks $1 \ldots t$. Using data $D_t \setminus C_t$ and having prior $M^{(t-1)}$, new model posterior $M^t$ is learnt. For predictive purposes at this stage (the test data comes from tasks $1 \ldots t$), a new posterior $M^t_{pred}$ is learnt with $M^t$ as prior and with data $C_t$. Note that $M^t_{pred}$ is used only for predictions at this stage, and does not have any role in the subsequent learning of, say, $M^{(t+1)}$. Such a predictive model is learnt after every new task, and discarded thereafter. Intuitively it makes sense as some new learnt weights for future tasks can help the older task to perform better (backward transfer) at testing time.

Coreset selection can be done either through random selection or $K$-center greedy algorithm  Gonzalez (1985). Next, the posterior is decomposed as follows:

$$p(\theta|D_{1:t}) \propto p(\theta|D_{1:t} \setminus C_t) p(C_t|\theta) \approx \tilde{q}_t(\theta) p(C_t|\theta)$$

where, $q(\theta)$ is the variational posterior obtained using the current task training data, excluding the current coreset data. Applying this trick in a recursive fashion, we can write:

$$p(\theta|D_{1:t} \setminus C_t) = p(\theta|D_{1:t-1} \setminus C_{t-1}) p(D_t \cup C_{t-1} \setminus C_t|\theta) \approx \tilde{q}_{t-1}(\theta) p(D_t \cup C_{t-1} \setminus C_t|\theta)$$

We then approximate this posterior using variational approximation as $\tilde{q}_t(\theta) = proj(\tilde{q}_{t-1}(\theta) p(D_t \cup C_{t-1} \setminus C_t|\theta))$ Finally a projection step is performed using coreset data before prediction as follows: $q_t(\theta) = proj(\tilde{q}_t(\theta) p(C_t|\theta))$. This way of incorporating coresets into coreset data before prediction tries to mitigate any residual forgetting. Algorithm 1 summarizes the training procedure for our model for setting with known task boundaries.

## D  DYNAMIC EXPANSION METHOD

Although our inference scheme uses a truncation-based approach for the IBP posterior, it is possible to do inference in a truncation-free manner. One possibility is to greedily grow the layer width until performance saturates. However we found that this leads to a bad optima (low peaks of likelihood). We can leverage the fact that, given a sufficiently large number of columns, the last columns of the IBP matrix tends to be all zeros. So we increase the number of hidden nodes after every iteration to keep the number of such empty columns equal to a constant value $\mathcal{T}^l$ in following manner.

$$C_j^l = C_{j+1}^l \prod_i^{D^l} \mathbb{I}(B_{ij}^l = 0), \quad G^l = \mathcal{T}^l - \sum_{j=1}^{K^l} C_j^l \tag{12}$$

where $l$ represents current layer index, $B^l$ is the sampled IBP mask for current task, $C_j^l$ indicates if all columns from $j^{th}$ column onward are empty. $G^l$ is the number of hidden units to expand in the current network layer.

## E  OTHER PRACTICAL DETAILS

### E.1  SEGREGATING THE HEAD

It has been shown in prior work on supervised continual learning Zeno et al. (2018) that using separate last layers (commonly referred to as "heads") for different tasks dramatically improves

---

**Algorithm 1** Nonparametric Bayesian CL

---

**Input:** Initial Prior $p_0(\Theta)$
Initialize the network parameters and coresets
Initialize : $p_{\text{new}} \leftarrow p_0(\Theta)$
**for** $i = 1$ **to** $T$ **do**
    Observe current task data $D_t$;
    Update coresets (Sec. C);
    **Masked Training**;
        $\mathcal{L}_t \leftarrow$ ELBO with prior $p_{\text{new}}$;
        $\Theta_t \leftarrow \arg\min \mathcal{L}_t$;
    **Selective Finetuning**;
        Fix the IBP parameters and learned mask;
        $\Theta_t \leftarrow \arg\min \mathcal{L}_t$;
    $p_{\text{new}} \leftarrow q_t(\Theta)$;
    $p_{\text{new}} \leftarrow \text{Mask}(p_{\text{new}})$ using Eq 11;
    Perform prediction for given test set..
**end for**

---

performance in continual learning. Therefore, in the supervised setting, we use a generalized linear model that uses the embeddings from the last hidden layer, with the parameters up to the last layer involved in transfer and adaptation. Although we do report comparision of single head models available in Sec H.2.

### E.2 SPACE COMPLEXITY

The proposed scheme entails storing a binary matrix for each layer of each task which results into 1 bit per weight parameter, which is not very prohibitive and can be efficiently stored as sparse matrices. Moreover, the tasks make use of very limited number of columns of the IBP matrix, and hence does not pose any significant overhead. Space complexity grows logarithmically with number of tasks $T$ as $\mathcal{O}(M + T \log_2(M))$ where M number of parameters.

### E.3 ADJUSTING BIAS TERMS

The IBP selection acts on the weight matrix only. For the hidden nodes *not* selected in a task, their corresponding biases need to be removed as well. In principle, the bias vector for a hidden layer should be multiplied by a binary vector $\boldsymbol{u}$, with $u_i = \mathbb{I}[\exists d : B_{d,i} = 1]$. In practice, we simply scale each bias component by the maximum reparameterized Bernoulli value in that column.

### E.4 SELECTIVE FINETUNING

While training with reparameterization (Gumbel-softmax), the sampled masks are close to binary but not completely binary which reduces performance a bit with complete binary mask. So we fine-tune the network with fixed masks to restore performance. A summarized version of Algorithm 1 summarizes our models training procedure. The method for update of coresets that we used are similar to as it was proposed in Nguyen et al. (2018).

## F ADDITIONAL INFERENCE DETAILS

**Sampling Methods** We obtain unbiased reparameterized gradients for all the parameters of the variational posterior distributions. For the Bernoulli distributed variables, we employ the Gumbel-softmax trick Jang et al. (2017), also known as CONCRETE Maddison et al. (2017). For Beta distributed $v$'s, the Kumaraswamy Reparameterization Gradient technique Nalisnick & Smyth (2017) is used. For the real-valued weights, the standard location-scale trick of Gaussians is used.

Inference over parameters $\phi$ that involves a random or stochastic node $Z$ (i.e $Z \sim q_\phi(Z)$) cannot be done in a straightforward way, if the objective involves Monte Carlo expectation with respect that random variable ($\mathcal{L} = \mathbb{E}_{q_\phi z}(L(z))$). This is due to the inability to back-propagate through a

random node. To overcome this issue, Kingma & Welling (2013) introduced the reparametrization trick. This involves deterministically mapping the random variable $Z = f(\phi, \epsilon)$ to rewrite the expectation in terms of new random variable $\epsilon$, where $\epsilon$ is now randomly sampled instead of $Z$ (i.e $\mathcal{L} = \mathbb{E}_{q\epsilon}[L(\epsilon, \phi)]$). In this section, we discuss some of the reparameterization tricks we used.

### F.1 GAUSSIAN DISTRIBUTION REPARAMETERIZATION

The weights of our Bayesian nueral network are assumed to be distributed according to a Gaussian with diagonal variances (i.e $V_k \sim \mathcal{N}(V_k | \mu_{V_k}, \sigma^2_{V_k})$). We reparameterize our parameters using location-scale trick as:

$$V_k = \mu_{V_k} + \sigma_{V_k} \times \epsilon, \quad \epsilon \sim \mathcal{N}(0, I)$$

where $k$ is the index of parameter that we are sampling. Now, with this reparameterization, the gradients over $\mu_{V_k}, \sigma_{V_k}$ can be calculated using back-propagation.

### F.2 BETA DISTRIBUTION REPARAMETERIZATION

The beta distribution for parameters $\nu$ in the IBP posterior can be reparameterized using Kumaraswamy distribution Nalisnick & Smyth (2017), since Kumaraswamy distribution and beta distribution are identical if any one of rate or shape parameters are set to 1. The Kumaraswamy distribution is defined as $p(\nu; \alpha, \beta) = \alpha\beta\nu^{\alpha-1}(1 - \nu^\alpha)^{\beta-1}$ which can be reparameterized as:

$$\nu = (1 - u^{1/\beta})^{1/\alpha}, \quad u \sim U(0, 1)$$

where $U$ represents a uniform distribution. The KL-Divergence between Kumaraswamy and beta distributions can be written as:

$$KL(q(\nu; a, b) || p(\nu; \alpha, \beta)) = \frac{a - \alpha}{a} \left( -\gamma - \Psi(b) - \frac{1}{b} \right) + \log ab + \log(B(\alpha, \beta)) - \frac{b}{1 - b}$$

$$+ (\beta - 1)b \sum_{m=1}^{\infty} \frac{1}{m + ab} B(\frac{m}{a}, b) \tag{13}$$

where $\gamma$ is the Euler constant, $\Psi$ is the digamma function and B is the beta function. As described in Nalisnick & Smyth (2017), we can approximate the infinite sum in Eq.13 with a finite sum using first 11 terms.

### F.3 BERNOULLI DISTRIBUTION REPARAMETERIZATION

For Bernoulli distribution over mask in the IBP posterior, we employ the continuous relaxation of discrete distribution as proposed in Categorical reparameterization with Gumbel-softmax Jang et al. (2017), also known as the CONCRETE Maddison et al. (2017) distribution. We sample a concrete random variable from the probability simplex as follows:

$$B_k = \frac{\exp((\log(\alpha_k) + g_k)/\lambda)}{\sum_{i=1}^{K} \exp((\log(\alpha_i) + g_i)/\lambda)}, \quad g_k \sim G(0, 1)$$

where, $\lambda \in (0, \infty)$ is a temperature hyper-parameter, $\alpha_k$ is posterior parameter representing the discrete class probability for $k^{th}$ class and $g_k$ is a random sample from Gumbel distribution $G$. For binary concrete variables, the sampling reduces to the following form:

$$Y_k = \frac{\log(\alpha_k) + \log(u_k/(1 - u_k))}{\lambda}, \quad u \sim U(0, 1)$$

then, $B_k = \sigma(Y_k)$ where $\sigma$ is sigmoid function and $u_k$ is sample from uniform distribution U. To guarantee a lower bound on the ELBO, both prior and posterior Bernoulli distribution needs to be replaced by concrete distributions. Then the KL-Divergence can be calculated as difference of log density of both distributions. The log density of concrete distribution is given by:

$$\log q(B_k; \alpha, \lambda) = \log(\lambda) - \lambda Y_k + \log \alpha_k - 2\log(1 + \exp(-\lambda Y_k + \log \alpha_k))$$

With all reparameterization techniques discussed above, we use Monte Carlo sampling for approximating the ELBO with sample size of 10 while training and a sample size of 100 while at test time.

## G  IBP HYPERPARAMETER $\alpha$

In this section, we discuss the approach to tune the IBP prior hyperparameter $\alpha$. We found that using a sufficiently large value of $\alpha$ without tuning performs reasonably well in practice. However, we experimented with other alternatives as well. For example, we tried adapting $\alpha$ with respect to previous posterior as $\alpha = max(\alpha, max(a_\nu))$ for each layer, where $a_\nu$ is Beta posterior shape parameter. Several other considerations can also be made regarding its choice.

### G.1  SCHEDULING ACROSS TASKS

Intuitively, $\alpha$ should be incremented for every new task according to some schedule. Information about task relatedness can be helpful in formulating the schedule. Smaller increments of $\alpha$ discourages creation of new nodes and encourages more sharing of already existing connections across tasks.

### G.2  LEARNING $\alpha$

Although not investigated in this work, one viable alternative to choosing $\alpha$ by cross-validation could be to learn it. This can be accommodated into our variational framework by imposing a gamma prior on $\alpha$ and using a suitably parameterized gamma variational posterior. The only difference in the objective would be in the KL terms: the KL divergence of $v$ will then also have to estimated by Monte Carlo approximation (because of dependency on $\alpha$ in the prior). Also, since gamma distribution does not have an analytic closed form KL divergence, the Weibull distribution can be a suitable alternative Zhang et al. (2018).

## H  ADDITIONAL RESULTS: SUPERVISED CONTINUAL LEARNING

In this section, we provide some additional experimental results for supervised continual learning setup. Table 2 shows final mean accuracies over 5 tasks with deviations, obtained by all the approaches on various datasets. It also shows that our model performs comparably or better than the baselines. We have included some more models in this comparison namely, HIBNN (Kessler et al., 2020), UCL (Ahn et al., 2019), HAT (Serrà et al., 2018) and A-GEM (Chaudhry et al., 2019). Note that coreset based replay is not helping much in our case, In of VCL use of coresets performs better since it forces all parameters to be shared leading to catastrophic forgetting. Our method has very less catastrophic forgetting hence the use of coresets does not improve performance significantly. Although in cases where we do not grow the model size dynamically and keep feeding tasks to it even after the model has reached its capacity (model will be forced to share more parameters), it will lead to forgetting and their use of coresets might help as it did for VCL.

### H.1  LEARNED NETWORK STRUCTURES

In this section, we analyse the network structures that were learned after training our model. As

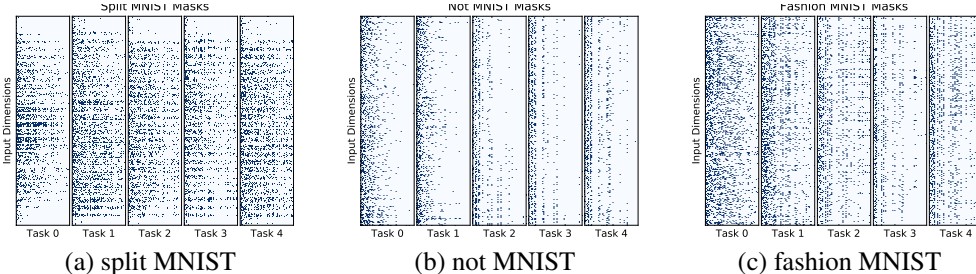

(a) split MNIST          (b) not MNIST          (c) fashion MNIST

Figure 6: Learned masks for input to first hidden layer weights on split MNIST(**left**), not MNIST(**middle**) and fashion MNIST(**right**) datasets. Darker color represent active weights.

we can see in Fig. 6(a), the masks are captured on the pixel values where the digits in MNIST

| Method | s-MNIST | n-MNIST | p-MNIST | f-MNIST | s-Cifar100 |
|---|---|---|---|---|---|
| Naïve | 79.615±0.7 | 72.339±0.8 | 90.090±0.4 | 79.319±0.6 | 47.082±0.7 |
| Rehearsal | 99.102±0.3 | 95.203±0.5 | 97.565±0.3 | 97.981±0.3 | 61.110±0.4 |
| EwC | 81.530±0.4 | 90.297±0.6 | 95.392±0.5 | 86.577±0.4 | 55.157±0.2 |
| IMM (mode) | 92.206±0.6 | 84.442±0.4 | 96.433±0.5 | 88.765±0.4 | - |
| HIBNN | 98.712±0.4 | - | 97.003±0.3 | - | - |
| VCL | 98.952±0.3 | 93.732±0.3 | 97.353±0.3 | 97.970±0.2 | 63.994 |
| VCL(coreset) | 98.731±0.4 | 94.993±0.2 | 97.464±0.3 | 98.154±0.3 | - |
| A-GEM | - | - | 95.645±0.2 | - | 62.945±0.1 |
| HAT | 99.701 | 96.749 | 97.912 | 98.592 | 66.410 |
| DEN | 99.779±0.1 | 96.485±0.3 | 97.945±0.2 | 98.580±0.3 | - |
| UCL | 99.791 | 97.112 | 97.883 | 98.896 | 64.32 |
| RCL | 99.768±0.1 | 96.722±0.2 | 98.005±0.2 | 98.698±0.2 | 64.814±0.1 |
| Ours | 99.819±0.1 | **97.152**±0.2 | **98.180**±0.2 | 98.986±0.2 | **70.105**±0.2 |
| Ours(coreset) | **99.834**±0.1 | 97.061±0.2 | 98.163±0.3 | **98.990**±0.2 | 69.459±0.2 |

Table 2: Comparison of final mean accuracies on test set obtained using different methods over 10 runs except for some with zero deviations (1-2 runs). Deviations are rounded to 1 decimal place, very small deviations are kept as 0.1.

datasets have high value and zeros elsewhere which represents that our models adapts with respect to data complexity and only uses those weights that are required for the task. Due to the use of the IBP prior, the number of active weights tends to shrink towards the first few nodes of the first hidden layer. This observation enforces that our idea of using IBP prior to learn the model structure based on data complexity is indeed working. Similar behaviour can be seen in notMNIST and fashionMNIST in Fig. 6(b and c).

On the other hand Fig 7 (left) shows the sharing of weights between subsequent tasks of different datasets. It can be observed that the tasks that are similar at input level of representation have more overlapping/sharing of parameters (e.g split MNIST) in comparison to those that are not very similar (e.g permuted MNIST). It also shows Fig 7 (right) that the amount of total network capacity used by our model differs for each task, which shows that complex tasks require more parameters as compared to easy tasks. Since the network size is fixed, the amount of network usage for all previous tasks tends to converge towards 100 percent. This promotes parameter sharing but also introduces forgetting, since the network is forced to share parameters and is not able to learn new nodes.

## H.2  ADDITIONAL PERMUTED MNIST RESULT

We have done our experiments with separate heads for each task of permuted MNIST. Some approaches use a single head for permuted MNIST task and don't task labels at test-time. Here we compare some of the baselines (that supports single head) with our model (single head) on Permuted MNIST for 10 tasks. We also report number of epochs and average time to run for a rough comparision of time complexity taken by each model. To justify the choice of single hidden layer with

| Method | Epochs/Task | Time/Task (sec) | Avg acc(10 tasks) |
|---|---|---|---|
| Ours | 10 | 142 | 0.9794 |
| VCL | 100 | 380 | 0.9487 |
| EwC | 10 | 51 | 0.9173 |

200 units in MNIST like experiments, we compare our model on Permuted MNIST experiment with multiple network depths and with separate heads, From table 3 we can conclude that a single hidden layer is sufficient for obtaining good enough results. Further, to analyse the performance decrease and generality of approach with number of tasks, we perform Permuted MNIST experiment with separate heads and a single hidden layer of 200 units for different number of tasks. Table 4 shows

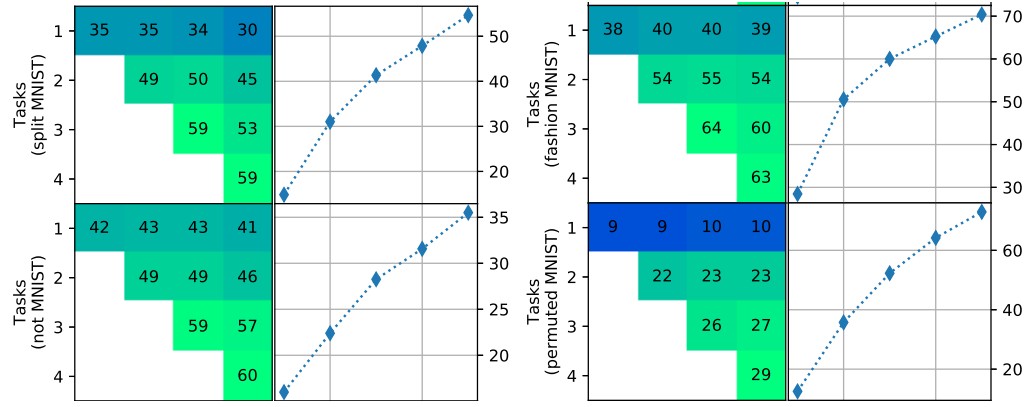

Figure 7: Percentage weight sharing between tasks (**left**), percentage of network capacity already used by previous tasks(**right**).

| Network hidden layer sizes | Avg accuracy (5 tasks) |
|---|---|
| [200] | $98.180 \pm 0.187$ |
| [100, 50] | $98.188 \pm 0.163$ |
| [250, 100, 50] | $98.096 \pm 0.152$ |

Table 3: Comparing performance on Permuted MNIST under different network configurations

that model quite stable and performance does not drop alot even with large number of tasks for a fixed model size.

| (Permuted MNIST) No. of tasks | Avg accuracy obtained |
|---|---|
| 5 | 98.180 |
| 10 | 98.062 |
| 20 | 97.874 |

Table 4: Comparison of model performance over different number of tasks for Permuted MNIST experiment

## H.3 ADDITIONAL CIFAR RESULT

MNIST data experiments are relatively easier to model and an approach might not generalize to more complex datasets like image or textual data. This section includes extra results on cifar-10 and cifar-100 datasets with comparisons to some very strong baselines for observing performance under complex settings.

| Method | split Cifar-100 (10 tasks) | split Cifar-100 (20 tasks) | split Cifar-10 (5-tasks) |
|---|---|---|---|
| VCL | 63.994 | 68.398 | 81.161 |
| HAT | 67.410 | 72.851 | 89.356 |
| Ours | 70.105 | 73.534 | 89.021 |

Table 5: Avg. accuracy obtained after all tasks are obtained on cifar-10 and cifar-100 datasets

Table 5 shows that our approach is comparable to the some strong baselines like HAT, VCL on complex tasks like cifar-10 and cifar-100 classifications. Therefore, suggesting that it can be generalized to more complex task settings. For split cifar-100 (20 tasks) each task is a 5 class classification task and, split cifar-10 has 2 class classification tasks.

## H.4 OTHER METRICS

We quantified and observed the forward and backward transfer of our and VCL model, using the three metrics given in Díaz-Rodríguez et al. (2018) on Permuted MNIST dataset as follows:

ACCURACY is defined as the overall model performance averaged over all the task pairs as follows:

$$Acc = \frac{\sum_{i \geq j} R_{i,j}}{\frac{N(N-1)}{2}}$$

where, $R_{i,j}$ is obtained test classification accuracy of the model on task $t_j$ after observing the last sample from task $t_i$.

FORWARD TRANSFER is the ability of previously learnt task to perform on new task better and is give by:

$$FWT = \frac{\sum_{i<j}^{N} R_{i,j}}{\frac{N(N-1)}{2}}$$

BACKWARD TRANSFER is the ability of newly learned task to affect the performance of previous tasks. It can be defined as:

$$BWT = \frac{\sum_{i=2}^{N} \sum_{j=1}^{i-1} (R_{i,j} - R_{j,j})}{\frac{N(N-1)}{2}}$$

We compare our model with VCL and other baselines over these three metrics in Table 6.

| Method | Accuracy | FWT | BWT |
|---|---|---|---|
| Naive | 90.090 | 0.1 | $-3.60e^{-2}$ |
| EwC | 95.392 | 0.1 | $-1.90e^{-2}$ |
| Rehearsal | 97.565 | 0.1 | $+1.30e^{-4}$ |
| VCL | 97.353 | 0.1 | $-4.00e^{-3}$ |
| Ours | 98.180 | 0.1 | $+1.33e^{-5}$ |

Table 6: Comparison on other metrics for permuted MNIST dataset

We can observe that backward transfer for our model is more as compared to most baselines, which shows that our approach has suffers from less forgetting as well. On the other hand forward transfer seems to give close to random accuracy (0.1) which is due to the fact that the model is not trained on the correct class labels and is asked to predict the correct label. So this metric is not very useful here; an alternative would be to train a linear classifier on the representations that are learned after each subsequent tasks for future task.

## I UNSUPERVISED CONTINUAL LEARNING

Here we describe the complete generative model for our unsupervised continual learning approach. The generative story for unsupervised setting can be written as follows (for brevity we have omitted the task id $t$):

$$\boldsymbol{B}^l \sim IBP(\alpha)$$
$$\boldsymbol{V}^l_{d,k} \sim \mathcal{N}(0, \sigma_0^2)$$
$$\boldsymbol{W}^l = \boldsymbol{B}^l \odot \boldsymbol{V}^l$$
$$\boldsymbol{W}^{out}_{d,k} \sim \mathcal{N}(0, \sigma_0^2)$$
$$\boldsymbol{Z}_n \sim \mathcal{N}(\mu_z, \sigma_z^2)$$
$$\boldsymbol{X}_n \sim Bernoulli(\sigma(\boldsymbol{W}^{out}\phi_{NN}(\boldsymbol{W}, \boldsymbol{Z}_n)))$$

where, $\mu_z, \sigma_z^2$ are prior parameters of latent representation; they can either be fixed or learned, and $\sigma$ is the sigmoid function. The stick-breaking process for the IBP prior remains the same here as well. For doing inference here, once again we resort to structured mean-field assumption:

$$q(\mathbf{Z}, \mathbf{V}, \mathbf{B}, \mathbf{v}) = q(\mathbf{Z}|\mathbf{B}, \mathbf{V}, \boldsymbol{\nu}, \mathbf{X})q(\mathbf{V})q(\mathbf{B}|v)q(\mathbf{v})$$

where, $q(\mathbf{Z}|\mathbf{B}, \mathbf{V}, \boldsymbol{\nu}, \mathbf{X}) = \prod_{n=1}^{N} \mathcal{N}(\mu_{\phi_{NN}}, \sigma_{\phi_{NN}}^2)$, and $\phi_{NN}$ is IBP masked neural network used for amortization of Gaussian posterior parameters. Rest of variational posteriors are factorized in a similar way as in the supervised approach. Evidence lower bound calculation can done as explained in section 3.3.

### I.1 ADDITIONAL EXPERIMENTAL RESULTS FOR UNSUPERVISED CONTINUAL LEARNING

In this section, we show further results for unsupervised continual learning. Fig 10 shows, for MNIST and notMNIST datasets, how the likelihoods vary for individual tasks as subsequent tasks arrive. It can be observed that the individual task accuracies learned by our model are better than

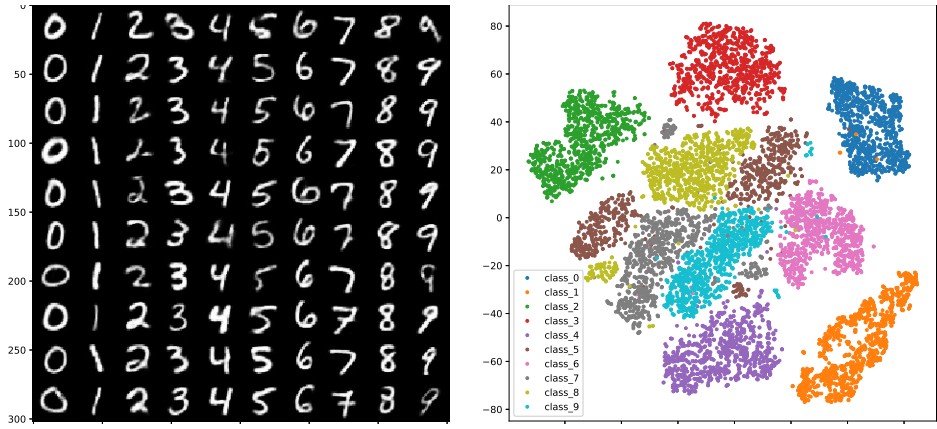

Figure 8: On MNIST dataset (**left**) Reconstruction of images after all tasks have been observed. (**right**) t-SNE plot of each class after all tasks have been observed.

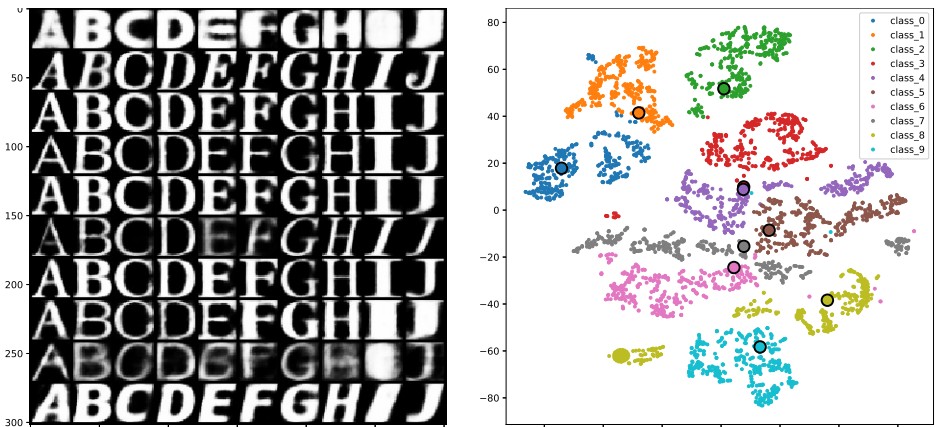

Figure 9: On notMNIST dataset (**left**) Reconstruction of images after all tasks have been observed. (**right**) t-SNE plot of each class after all tasks have been observed.

other baselines; this suggests that use of new weights when needed helps in retaining a better optima per task, and also the deterioration of our model is much less as compared to other model, representing effective protection against catastrophic forgetting. Fig 12 shows the reconstructed images of MNIST and also the t-SNE plot of latent codes our model produces. it can be observed that reconstruction quality is good despite heavy constraints on the model. Fig 11a shows the generated

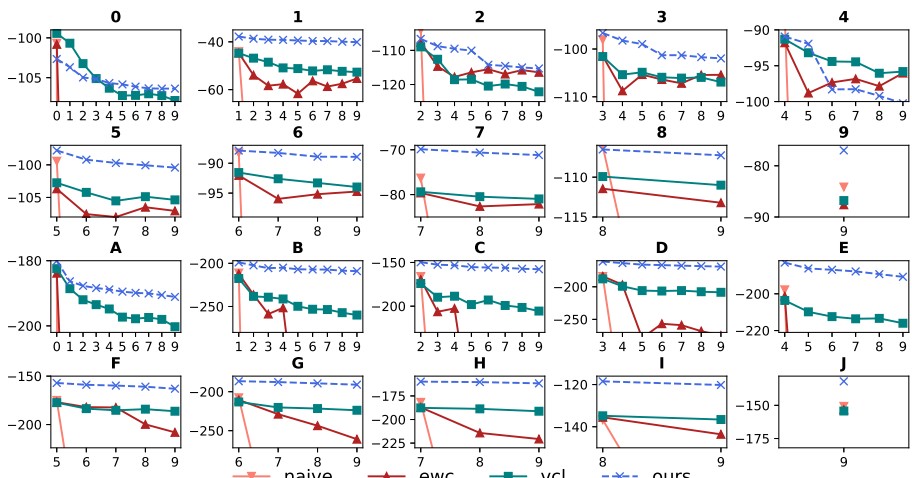

Figure 10: Generative Model : Test likelihood decays of individual tasks after subsequent tasks have been observed. (Top two) represents MNIST and (Bottom two) represents notMnist datasets.

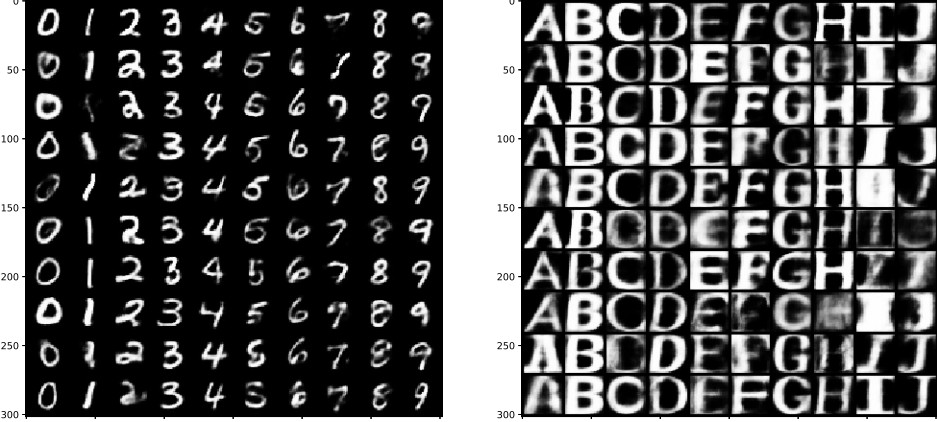

(a) Generated samples on MNIST dataset after all tasks have been observed

(b) Generated samples on notMNIST dataset after all tasks have been observed

Figure 11: Generated Samples

samples from the learned prior over latent space after all tasks are observed.

Similarily, Fig 9 shows the reconstructed images of not MNIST dataset and the t-SNE plot of latent codes our model produces, and Fig 11b shows the generated samples from the learned prior over latent space after all tasks are observed.

| Benchmarks | MNIST (error) | | | not MNIST (error) | | |
|---|---|---|---|---|---|---|
| | 3-KNN | 5-KNN | 10-KNN | 3-KNN | 5-KNN | 10-KNN |
| Naive | 30.1% | 33.1% | 36.0% | 20.6% | 24.87% | 30.8% |
| EwC | 16.6% | 19.48% | 22.3% | 11.7% | 13.1% | 17.8% |
| VCL | 17.0% | 19.02% | 30.2% | 12.3% | 13.8% | 16.5% |
| Ours | **0.37**% | 0.40% | 0.51% | **0.08**% | 0.09% | 0.21% |

Table 7: Unsupervised learning benchmark comparison with sampled latents using multiple K-nearest neighbour errors obtained from each baseline

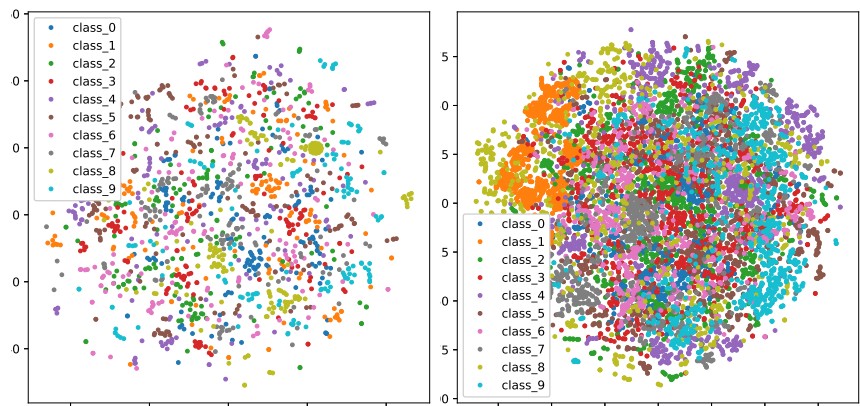

Figure 12: t-SNE plot of latent space of VCL model on notMNIST (left) and MNIST (right) datasets

REPRESENTATION LEARNING   In t-SNE plots, it can be observed that the latent space for MNIST dataset is more clearly seperated as compared to notMNIST dataset. This can be attributed to the abundance of data and less variation in MNIST dataset as compared to notMNIST dataset.  we further analyzed the representations that were learned by our model by doing $K$-Nearest Neighbour classification on the latent space.  Table 7 shows the KNN test error of our model and few other benchmarks on MNIST and notMNIST datasets. We performed the test with three different values for $K$. As shown in the table, the representations learned by other baselines are not very useful (as evidenced by the large test errors), since the latent space are not shared among the tasks, whereas our model uses a shared latent space (yet modulated for each task based on the learned task-specific mask) which results in effective latent representation learning.

## J   TASK AGNOSTIC SETTING

We extended our unsupervised continual learning model to a generative mixture model, where each mixture component is considered as a task distribution (i.e $p(\boldsymbol{X}) = \sum_{k=1}^{K} p(\boldsymbol{X}|\boldsymbol{t}=k)p(\boldsymbol{t}=k)$ with $\boldsymbol{t}$ representing the task identity). Here, $p(\boldsymbol{t}=k)$ can be assumed to be a uniform distribution but it fails to consider the degree upto which each mixture is being used. Therefore, we keep a count over the number of instances belonging to each task and use that as prior (i.e $p(\boldsymbol{t}=k) = \frac{N_k}{N}$, with $N_k$ being effective number of instances belonging to task $k$ and $N = \sum_k N_k$).

DETECTING BOUNDARIES   Inspired from Rao et al. (2019a), we rely on a threshold to determine if the data point is an instance from a new task or not. During training, any instance with $\mathbb{E}_{p(\boldsymbol{t}_n|\boldsymbol{x}_n)}(\text{ELBO}_{\boldsymbol{t}_n})$ less than threshold $T_{\text{new}}$ is added to a buffer $\mathcal{D}_{\text{new}}$. Once the buffer $\mathcal{D}_{\text{new}}$ reaches

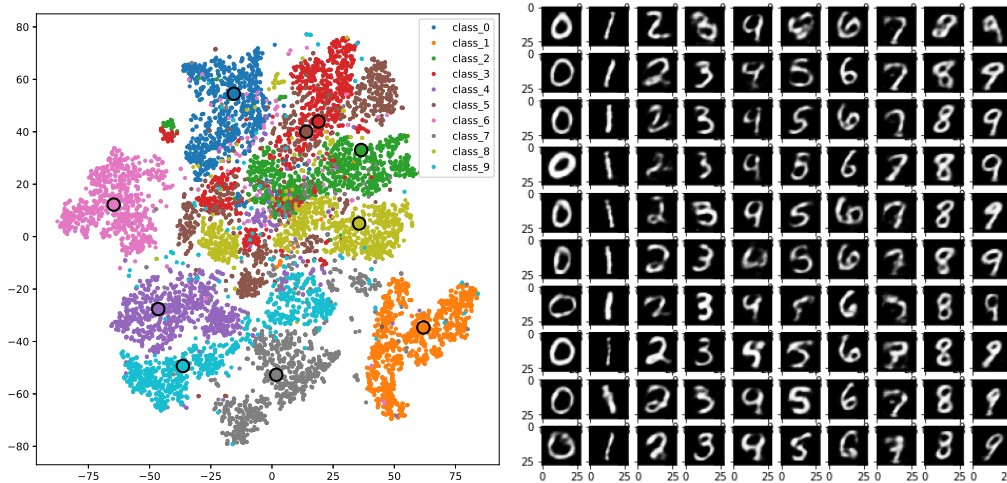

Figure 13: Reconstructed MNIST samples and T-SNE plots of our task agnostic setting

a fixed size limit $M$, we extend our network with new task parameters and train our network on $\mathcal{D}_{\text{new}}$, with known task labels (i.e $p(y = T + 1) = 1$ where $T$ is total number of tasks learned)

SELECTIVE TRAINING    Note that training this mixture model will require us to have all task specific variational parameters to be present at every time step unlike the case in earlier settings where we only need to store the masks and can discard the variational parameters of previously seen tasks. This will result in storage problems since the number of parameters will grow linearly with the number of tasks. To overcome this issue we fix the task specific mask parameters and prior parameters before the network is trained on new task instances. After the task specific parameters have been fixed, the arrival of data belonging to a previously seen task $t_{prev}$ is handled by training the network parameters with task specific masks $B_{prev}$.

REPRESENTATION LEARNING    It makes more sense do learn representations when we don't have target class labels or task labels. As discussed, we trained our model using a gating mechanism with a threshold value of $-130$. Fig 13 qualitatively shows the t-SNE plots and reconstruction for each class data points. Based on these results, we can conclude that the task boundaries are well understood and separated by our model.

