# OpenReview forum: "A Unified Bayesian Framework for Discriminative and Generative Continual Learning"
_ICLR.cc/2021/Conference — Reject_

### Official Review · AnonReviewer2 · 2020-10-19
**Automated probabilistic structure learning for CL.**

**Rating:** 7
**Confidence:** 4

**Review:**

The authors present a new structure-learning approach to Continual Learning, by modelling each hidden layer using a nonparametric Bayesian prior, a technique inspired by recent work on learning sparse NNs. Using this technique, a sparse subset of available weights is used for each task, selectively allowing knowledge sharing between subsequent task while reducing catastrophic forgetting on the deactivated connections.

This makes for an overall very convincing submission, reflected by my minor criticism. Well placed among other recent publications on CL in top venues.

Pros (in no particular order):
- The method is principled and follows naturally and elegantly in the VCL framework. The presentation follows a clear narrative that is easy to follow.
- The method comes with a task detection mechanism.
- Both discriminative and generative modelling are naturally supported in the same framework.
- The Appendix discusses all necessary details to a level well-above standard in the literature. Analysis covers most of the interesting questions that can naturally be asked about this method.
- Presented results are overall strong and cover standard evaluation in the literature. I was pleased to see experiments in an application other than supervised image classification (here unsupervised learning).
- A simple heuristic for dynamic expansion is introduced, a worthwhile direction for future research.

Cons (in no particular order):
- This method requires storage of a binary matrix for task and each layer in the network. However, the authors show that space complexity grows logarithmically with the number of tasks, which is likely to make this an acceptable trade-off.

Author feedback:
- IMHO the main contribution of this work is its ideas on structure learning in the context of Continual Learning. This is however not reflected in the title.
- The paper would benefit from clearly stating why structure learning for CL is a worthwhile direction to pursue. Well-written papers clearly state why the proposed direction is a worthwhile method of investigation.
- Figure 3 is missing axis labels

---

> ### Author Response · Authors · 2020-11-17
> **Author Response**
>
> Thank you very much for your very considerate and thoughtful review. We have modified the paper to incorporate your feedback. Any new feedback on the updated version is appreciated and will be incorporated in the final version as well. Below are the responses to your questions and comments.
>
> **1. On why the proposed direction is a worthwhile investigation.**
> We are very thankful that you appreciated our approach and also for pointing out our work's unique strengths in a very explicit way. Regarding your advice for clearly stating why the structure learning is a worthwhile investigation, we have added a discussion in paper (Page 2 first paragraph) explaining its importance. We will add structural learning in the title in the final version. Any other  feedback on this is appreciated and will be incorporated in the final version.
>
> We have updated the missing axis labels in figure 3.

---

> > ### Comment · AnonReviewer2 · 2020-11-23
> > **Reviewer Response**
> >
> > Thanks for your response. After reading the other authors' feedback, I remain convinced that the chosen score is appropriate.

---

### Official Review · AnonReviewer1 · 2020-10-20
**Interesting applicaton of Bayesian non-parametrics for continuous learning but the experiments could be more challenging**

**Rating:** 6
**Confidence:** 3

**Review:**

Summary:  The paper proposes a continual learning framework based on Bayesian non-parametric approach.  The hidden layer is modeled using Indian Buffet Process prior.  The inference uses a structured mean-field approximation with a Gaussian family for the weights, and Beta-Bernoulli for the task-masks.  The variational inference is done with Bayes-by-backprop on a common ELBO setup.  The experiments show less diminishing accuracy on the increment of tasks on five datasets for the discriminative problem, and for generation the methods learn one digit or character at a time on MNIST and notMNIST datasets.


Pros:
- The paper shows a structured mean-field approximation of the continual learning problem, and train a single hidden layer of 200 units on the discriminative and generative settings.
- The paper is easy to follow.

Cons:
- The structure of the network is rather shallow.  The authors also mentioned the challenge in their conclusion.
- The experiments were done on rather simple datasets.  The paper introduces a proof of concept instead of showing the capabilities of the proposal in complex scenarios.
- The mixture of the structured mean-field approximation seems straightforward and builds extensively on previous work.  The novelty may be on making the network train using this approach, but the proposal uses a rather simple layer configuration (one layer).

Comments:
- In Section 5.1 you mention that you report the mean accuracies on the different tasks (Fig. 2).  How many versions of the tasks do you generate and average over?  Adding error bars would help to see the variance of the methods over the tasks.

Minor comments:
- Typo P6 par2 "tries to adapts"
- Typo P7 par 3 "comapred"

Overall rating:
The idea of applying Bayesian non-parametric for continuous learning is interesting, and the authors show a simple implementation on simple datasets.  The evaluated tasks are extremely related and in a general continuous learning setup this method may not work.  More extensive and complex experiments (i.e., more complex databases and setups for discrimination as well as for generation) may shed some light on the process.  Due to all these issues I rate the paper as a 5.

---

> ### Author Response · Authors · 2020-11-17
> **Author Response**
>
> Thanks for your thoughtful review. Below is our response to your questions and comments. We have modified the paper to reflect your feedback, any feedback on the new version is appreciated. We hope our response below (along with the additional results we present below and in the revised manuscript) helps address your concerns and will hopefully help you reevaluate your assessment of our paper.
>
> **1. “The structure of the network is rather shallow”.**
> For MNIST datasets, we could have gone with deeper models, but as it can be seen in the table below, the single hidden layer is sufficient to achieve good enough results (10 runs each). We have added the analysis table in the appendix (Table 3).
>
> | Hidden layers  | $\ $ avg. acc +/- std |
> |----------------|----------------------|
> | [200]          | 98.180 +/- 0.187     |
> | [100, 50]      | 98.188 +/- 0.163     |
> | [250, 100, 50] | 98.096 +/- 0.152     |
>
> We would also like to highlight our results on split cifar-100 datasets (Fig. 2-5) for which we used an Alex-Net like structure with three convolutional layers of 128, 256, 512 channels with 4 × 4, 3 × 3, 2 × 2 channels followed by two dense layers of 2048, 2048 units each, as mentioned in Appendix Section B.1, which indicates that the proposed model can indeed be applied to deeper models.
>
> Your question regarding the depth also points towards a more general challenge that variational inference in deeper models might be a bit slow to train due to the sampling step in ELBO calculations, which is a trade-off generally done for using variational inference to attain benefits of a bayesian model over deterministic models (e.g., automatic parameter specific regularization via prior, obtaining a distribution over parameters and latent variables, uncertainty in predictions, generative capability, etc). Although we used a training sample size of 10 points and testing sample size of 100 points (Time analysis on permuted MNIST can be found in Appendix section H.2), we found that using a sample size of 4-5 for training and 10 for testing was enough for convolutional neural network to reach same level of performance as mentioned in paper. However, experiments for generalizing on very large and deep neural architectures can be a future work as you mention.
>
> **2. “The experiments were done on rather simple datasets”**
> We agree that the MNIST dataset is a rather simple dataset but, as justified in Anonreviewer3’s experimental evaluation, most related works consider these settings and is the reason why we used them as it also enables easier model validation and comparison. On the other hand, we would also like to highlight that Split CIFAR-100 dataset, which is much complex than MNIST, can definitely give more insights about the performance on complex data, also avoiding catastrophic forgetting while performing a sequential generative modelling with good latent representation learning capabilities of proposed approach (as discussed in Fig. 4 , Fig 5 and Table 1) are much more complex as compared to classification of MNIST digits which shows that given a proper network size it can also handle complex tasks.
>
> **3. “The mixture of structured mean-field approximation seems straightforward and builds extensively on previous work. The novelty may be in making the network train using this approach, but the proposal uses a rather simple configuration (One Layer)”**
> While this is true, this is not the main focus of our work. Regarding this, we would also like to draw your attention towards some of the key/interesting aspects of our work that are pointed out by the other reviewers as well. In particular, (1) our method comes with a task detection mechanism, (2) both discriminative and generative modelling are naturally supported in the same framework (not generally available in most models), (3) the representation learning capability in the task-agnostic setting of our model is comparable to one of the SOTA baselines, (4) although not discussed in main pape, a simple heuristic for dynamically expanding (appendix D) the network without specifying any predefined size on which further research can be performed. Also our model is principled in terms of a rigorous Bayesian nonparametric approach and promotes further research in this direction.
>
> **4. “How many versions of the tasks do you generate and average over? Adding error bars would help to see the variance of the methods over the tasks”**
> We averaged over 10 runs for all our experiments; the variance information was moved to appendix due to space constraints and can be found in Appendix Table 2.
>
> In the revised version, we have corrected some of the typos mentioned by you (thanks for pointing those out) and some that we observed.

---

> > ### Comment · AnonReviewer1 · 2020-11-19
> > **I'm still concerned about the contribution and novelty**
> >
> > I thank the authors for their effort replying to my comments and updating the paper.
> >
> > However, I'm still not convinced on the added novelty that authors claim.  Moreover, the experiments, even though if they are performed on similar databases as the literature, are not convincing.  We should improve upon existing methods, instead of simply reproducing their limitations.  I urge the authors to think about improving further upon their experimental setup, and evaluating in a more rigorous setup.
> >
> > Due to these reasons, I stand by my original evaluation.

---

> > > ### Author Response · Authors · 2020-11-21
> > > **Thank you for your response**
> > >
> > > Thank you for your feedback. Please note that our experiments are not limited to just discriminative (supervised) setting but also generative (unsupervised) setting, task id inference, etc. Some of these capabilities are lacking in the best-performing methods. We have reported an extensive set of experiments on all of these settings (in the main paper as well as the supplementary material) and, as AnonReviewer2 points out, our analysis covers most of the interesting questions about such a method, and that the experimental evaluation is strong and extensive.
> > >
> > > That said, we appreciate your comments, following which we have tried another setup for classification tasks -- split notMNIST followed by splitMNIST followed by split fashionMNIST -- repeated under 3 different settings as Noise, Gaussian Blur, Background zigzag pattern, amounting to a total of 45 tasks with a model configuration of [input_dim, 200, 200, output_dim]. The results are as follows (will be added in the final version of the paper):
> > >
> > > Average accuracy obtained over 45 tasks: 97.24 %
> > >
> > > Accuracy variation of the first task (forgetting):
> > >
> > > 97.47, 97.47, 97.33, 97.33, 97.47, 97.47, 97.47, 97.33, 97.47, 97.47, 97.33, 97.47, 97.33, 97.33, 97.07, 96.8 , 96.93, 97.2 , 96.93, 97.2 , 97.2 , 97.2 , 97.2 , 97.07, 97.2 , 97.2 , 97.2 , 96.93, 96.8 , 97.07, 96.8 , 96.8 , 96.93, 96.8 , 96.67, 96.93, 96.93, 96.8 , 96.67, 96.67, 96.8 , 96.93, 96.93, 96.8 , 96.93
> > >
> > > The following table shows the average accuracy obtained for different configurations in this setup.
> > >
> > > | Avg. Accuracy      | Noise | Blurred | Background pattern |
> > > |--------------------|-------|---------|--------------------|
> > > | Split notMnist     | 97.1  | 96.41   | 95.89              |
> > > | Split Mnist        | 99.75 | 99.74   | 99.01              |
> > > | Split fashionMnist | 99.32 | 99.27   | 98.49              |
> > >
> > > Also, regarding novelty (which AnonReviewer2 and AnonReviewer3 also appreciate), we would like to highlight that use of IBP prior over network connections to perform structural learning hence fighting catastrophic forgetting is rather new and has not been tried so far to the best of our knowledge. The prior and approximate posterior is more general than VCL as in that it also considers the neural network structure as a random variable (prior and posterior). Other than that we would also like to highlight that along with supervised learning capability, our model has generative and in particular sequential representation learning ability which is far superior than existing baselines VCL, EwC etc. and most of the CL models are not intended for unsupervised setup. Also the extension to task-agnostic (unknown task boundaries) setting is not generally found in CL methods but it is more realistic than other setups with comparable representation learning capability to state-of-the-art methods in this domain. Unifying all this in a single approach while performing better or comparable is lacking in the current CL literature.
> > >
> > > With that, we would also like to highlight that our work can potentially also motivate interesting future work on CL using nonparametric Bayesian approaches. Other interesting extensions could be amortized parameter estimation and automated structure learning without needing to define the truncation level (not knowing the task complexity), scenarios where a model learns sequential generations but can separate them in latent space at the same time and if the data is non stationary could introduce Bayesian forgetting as discussed with reviewer3 and many other scenarios where bayesian modelling is more useful than deterministic modelling. Therefore, we believe our work has appropriate novelty and potential to fuel further research on continual learning.

---

### Official Review · AnonReviewer4 · 2020-10-28
**This paper proposed a Bayesian nonparametric approach to continual learning in the context of deep neural networks. It combines structure learning and online bayes. The framework uses the Indian Buffet Process and is an extension of the VCL work in (Nguyen et al. 2018). However, I don’t know which continual learning problem it is solving and thus it is hard for me to evaluate.**

**Rating:** 4
**Confidence:** 4

**Review:**

I found it difficult to evaluate this paper because the paper does not say which continual learning problem it is solving in the supervised learning case. Is it solving the class-incremental learning or task-incremental learning problem? Without knowing that, it is hard to make an assessment because the two problems are usually solved in very different ways and their evaluation protocols are different too. I tried to guess but get confused.

The paper writes in the evaluation section “To gauge the effectiveness of our model towards preventing catastrophic forgetting, we report (i) the test accuracy of first task after learning each of the subsequent tasks; and (ii) the average test accuracy over all previous tasks 1, 2, . . . t after learning each task.” And it also writes earlier “Omitting the task id t for brevity.” I am confused with these two statements. Do you need task id during training and testing? If you are solving the class-incremental learning, task id should not be used in training or testing, at least not testing. How do you do (i)? Do you only use the test instances of the first task? Do you restrict those instances to be classified into only the classes in the first task, or do you allow them to be classified to future classes in future tasks? For class-incremental learning, one should be getting the accuracy of all classes learned so far rather than each task. So, I am guessing that you are doing task-incremental learning. (ii) also gives me the same impression. If that is the case, the following systems are expected to be compared: Uncertainty-based Continual Learning with Adaptive Regularization (NIPS-2019) and Overcoming Catastrophic Forgetting with Hard Attention to the Task (Serrà et al., 2018). IBP is related to the mechanisms in (Serrà et al., 2018) and (Adel et al. 2020). It is desirable to have them compared.

If you are doing class-incremental learning, more recent baselines should be compared. The baselines used in your experiments are old.

Learning a Unified Classifier Incrementally via Rebalancing. CVPR 2019.
Overcoming catastrophic forgetting for continual learning via model adaptation. ICLR, 2019.
Large scale incremental learning. CVPR 2019
Random path selection for continual learning. NeurIPS 2019
Continuous learning of context-dependent processing in neural networks. Nature Machine Intelligence, 2019.
itaml : An incremental task-agnostic meta-learning approach, CVPR, 2020
Continual learning with hypernetworks. ICLR, 2020

In the experiment, varying the number of tasks for each dataset is also desired to show the generality of the proposed approach.

---

> ### Author Response · Authors · 2020-11-17
> **Author Response**
>
> Thank you very much for your review. Below, we respond to your questions and concerns. We have revised the paper to incorporate your feedback and to clarify the points that could have led to a possible confusion (in particular, class-incremental vs task-incremental). Any new feedback on the updated version is appreciated and will be incorporated in the final version as well.
>
> **1. “the paper does not say which continual learning problem it is solving in the supervised learning case”.**
> We apologize for the confusion but your guess based on our experimental setup is correct -- we have considered task-incremental learning in the supervised learning case. We have updated the paper with a more specific description (in Section 3, 1st paragraph and in conclusion) that, in this work, our supervised learning setup is based on task-incremental learning.
>
> **2. “Omitting the task id t for brevity”**
> In sec 3.1, we have updated the sentence to make it more clear that the generative story is for a single task in task-incremental setup.
>
> **3. “Do you need task id during training and testing”**
> In the supervised setting, we do need them. However, in the unsupervised setting, we have discussed a task-agnostic setting where we do not need the task identities during training or testing. Also, we believe that our task-agnostic setup can be used in scenarios of class-incremental learning where each class generation can be considered as a single task, and based on p( t = k | x) (probability of a  sample belonging to a task t), we can classify the test inputs. This can be explored further in future works, and we have mentioned this in the Conclusion section.
>
> **4. “systems that are expected to be compared”**
> In the revision, we have added comparisons with some of the suggested methods in Appendix Table 2, and will be happy to add more comparisons in the final version. That said, we would also like to highlight that we have already compared with some very strong and fairly recent CL baselines, such as DEN and RCL, which we believe should be considered as good baselines for judging our models performance. We would also like to point out that most of these methods do not generalize to unsupervised and task-agnostic settings, whereas our work generalizes to such scenarios with ease as discussed in the paper due to its principled Bayesian non-parametric framework.
>
> **5. “varying the number of tasks for each dataset is also desired to show the generality of the proposed approach”**
> We understand your concern and have conducted additional experiments (please see below) which we have added in the revised version. For permutedMNIST, we have reported results on 10-tasks in Appendix H.2, for split data experiments since the number of classes are fixed, the number of tasks that we considered are limited to 5 in MNIST like datasets and 10 in Split CIFARr100 experiment.
>
> | # Tasks | $\ $Avg. acc |
> |---------|----------|
> | 5-task  | 98.180   |
> | 10-task | 98.062   |
> | 20-task | 97.874   |
>
> For comparison, on permuted MNIST experiment, average accuracy with varying number of tasks for a single hidden layer network with [200] units as shown above has been added in Appendix (Table 4).

---

> > ### Comment · AnonReviewer4 · 2020-11-21
> > **I am still unsatisfied with the response**
> >
> > Thanks for classifying that for supervised learning you do task-incremental learning. However, one important comment is not addressed satisfactorily. I checked Table 2, you compared only HAT and DEN on MNIST and f-MNIST, which are easy problems, and all top algorithms do equally well. But there are no results for CIFAR100. In Figure 2 in the paper, there are no results for DEN on CIFAR100 either although some other plots have DEN’s results. Can you please compare your method with HAT and DEN using CIFAR100 - 10 and 20 tasks, and CIFAR10 – 5 tasks, which can better set different algorithms apart? HAT is perhaps the best algorithm on average for task-incremental learning. It has almost no catastrophic forgetting because it masks each task. Furthermore, HAT does not memorize any training data from any task, but your algorithm for supervised learning needs, which is a disadvantage compared to HAT and many others.
> > Many baselines that you have compared are not designed for task-incremental learning but for class-incremental learning, which is a much harder problem. If you don’t modify their code to enable them to work in the task-incremental learning setting, they obviously won’t do well. Without getting these issues resolved, I would like to keep my score.

---

> > > ### Author Response · Authors · 2020-11-24
> > > **Thank you for your response**
> > >
> > > Thanks for your response, we have provided a comparison with HAT and VCL in the table mentioned below (We have updated the paper and will add more comparisons in the final version) We could not produce results on DEN as its implementation on convolutional networks is not available.
> > >
> > > | Method | Cifar-100 (10 tasks) | Cifar-100 (20 tasks) | Cifar-10 (5 tasks) |
> > > |--------|----------------------|----------------------|--------------------|
> > > | VCL    | 63.994               | 68.398               | 81.161             |
> > > | HAT    | 67.410               | 72.851               | 89.106             |
> > > | Our    | 70.105               | 73.534               | 88.987             |
> > >
> > > Further, we want to clarify that our approach does not require memorization of any training data. If you are referring to the coreset method, we have mentioned in related works paragraph 2 that it's just an additional mechanism that can be used in variational bayes method but it is NOT a requirement for our method. Also, in appendix section H, it is discussed why use of coreset is not significant for our model’s performance. Also, many methods that we have compared with like Naive, RCL, DEN, EwC, VCL, UCL, Rehearsal etc. were developed for task-incremental learning. Permuted MNIST experiment by its nature is a task-incremental learning scenario and so for all baselines we have used their implementation that was developed for task-incremental learning scenarios.
> > >
> > > Finally, we would like to emphasize that comparing our approach with approaches such as HAT is only sensible for discriminative continual learning. Our approach is much more general and applicable to a lot of more challenging settings for which HAT has not been designed for, such as generative (unsupervised) continual learning, and task-id inference. Even if HAT outperforms/does comparably on some of the discriminative continual learning dataset, please note that our framework has a much broader applicability, as recognized by the other reviewers as well. Bayesian methods are natural for continual learning problems (as shown in the VCL paper) and our work further establishes their efficacy for continual learning problems. We request the review to take in consideration these aspects as well, rather than just comparison with HAT and other methods (which we do however provide as requested by the reviewer).

---

### Official Review · AnonReviewer3 · 2020-10-29
**Official Blind Review #3**

**Rating:** 8
**Confidence:** 5

**Review:**

This work proposes an online variational Bayesian (VB) approach to continual learning. The prior over neural network functions is both over the neural network structure and parameter values, where the structure is modelled by an Indian Buffet process (IBP) and the weights are drawn from a Gaussian.
Similarly, the approximate posterior is assumed to be IBP and factorised Gaussian as well and inference is performed through variational inference and reparametrization of the respective distributions.

The approach is similar to VCL in that it uses online VB for learning, however, the prior and approximate posterior is more general in that it also considers the neural network structure as a random variable (prior and posterior).

Theory:
The approach is theoretically sound and well motivated; the paper is presented well and easy to follow.
My main concern is that that the second paragraph of the paper motivates with scenarios where the ability to *adapt to dynamically changing environments or evolving data distributions* is essential. However, online VB assumes iid data. That is, the online algorithm should infer the posterior over all tasks rather than adapting to dynamically changing data distributions. Inference is sequentially, but the ordering of the task should in theory not matter - it only matters in practice as we perform approximations.
See e.g. [1] (ICLR 2020) for an approach that explicitly adapts (through forgetting) the distribution over neural network weights. It could be possible to extend this work to similar adaptation mechanisms, although for multi-task learning such adaptation/forgetting may not be desirable. I would appreciate a few comments on this and I think it should also be discussed shortly in the paper.

Experimental evaluation:
The experimental section considers scenarios that are very common in the CL literature. Unfortunately these are not the most interesting or insightful, as these are variants of MNIST. But since most related work considers these settings as well, the choice is justified.
The results are quite strong. I find the results on classification from the latent space of the unsupervised learning approach especially convincing and interesting (Table 1).

Related work:
The relation to [2] needs to be discussed in more detail. What exactly is the difference if the IPB is put on the activations rather than weights? What are pros and cons? I am aware that there are additional experiments in the supplementary material comparing to Kessler. Why do you think your approach outperformed the one of Kessler?

Another interesting aspect is that the coreset does not help much, which is in contrast to VCL. Do you think this is because the performance is already high? Or because the coreset selection algorithms (k-center, random) are unsuitable?

[1] Continual Learning with Bayesian Neural Networks for Non-Stationary Data, ICLR 2020
[2] Hierarchical indian buffet neural networks for bayesian continual learning

---

> ### Author Response · Authors · 2020-11-17
> **Author Response**
>
> Thank you very much for your interesting and thoughtful review. We have modified the paper to incorporate your feedback. Any new feedback on the updated version is appreciated and will be incorporated in the final version as well. Below are the responses to your questions and comments.
>
> **1. “Non-stationary data”**
> Thank you for highlighting a critical point that online VB assumes i.i.d. Data. Indeed, even our setup for continual learning can have a shift in data distribution and might not perform well in real world settings of continuously evolving distributional shifts. Since our approach is based on VCL, it can definitely be adapted to the problem setting [1] as suggested by you. In particular, the "Adaptation with Bayesian Forgetting" mentioned in [1] can be applied to our model to get a new KL divergence term between variational and true posterior obtained after applying Bayesian forgetting. However, for the task-agnostic setting the forgetting introduced should depend on the effective number of samples assigned to that task rather than the time delay. We have added a short discussion in the revised version (in the related works section last paragraph). Thanks again for bringing up this point.
>
> **2. “Comparison to Kessler et al”**
> Benefit of modelling weights rather than nodes is to achieve more flexibility in terms of structural learning. Our approach can learn sparse network connection between adjacent layer nodes with respect to a task, while Kessler et al’s approach learns sparsity in terms of active nodes and has to use all connections to that node from previous layer active nodes with respect to a given point. We have added the discussion (in related works section fourth paragraph)
>
> Reason for better performance of our model: The formulation of the approach itself is very different from Kessler et al. In their approach, each point selects a set of active nodes (matrix factorization) whereas, in our case, for a given task, each previous layer node selects next layer nodes to connect to (connection matrix) which gives us more flexibility in terms of modelling for a given task.
>
> Another key difference from Kessler et al is that (and as we mention in our paper) our approach can handle both discriminative and generative (unsupervised) continual learning whereas Kessler et al only consider discriminative learning.
>
> We would also like to point that, we couldn’t perform comparisons with their model in other settings such as unsupervised (as they did not extend it for unsupervised setting), task-agnostic setting is again not trivial as kessler’s approach does not have any task-inference mechanism.
>
> We would however like to point out that, modelling weights rather than nodes, requires more storage as we are storing mask parameters.
>
> **3. “Coresets not helping much”**
> Coreset based replay will help in the case of VCL since it forces all parameters to be shared, leading to catastrophic forgetting. We can explore other coreset selection methods as mentioned in [1] but we believe that our method performance is already high due to less catastrophic forgetting. Therefore, the use of coresets does not improve performance significantly. That said, in cases where we do not grow the model size dynamically and keep feeding tasks to it even after the model has reached its capacity (the model will be forced to share more parameters), it will lead to forgetting and in sich cases, the use of coresets might help as it did for VCL. We have added a discussion (in appendix H first paragraph), and the following experimental analysis on permuted MNIST confirms the observation.
>
> Additional experiment with a single layer of 10 hidden units for permuted MNIST (10 tasks).\
> Avg. accuracy without coresets:	93.01\
> Avg. accuracy with coresets:  		93.16
>
> [1] Continual Learning with Bayesian Neural Networks for Non-Stationary Data

---

> > ### Comment · AnonReviewer3 · 2020-11-20
> > **IBP on nodes vs all weights**
> >
> > Thanks for your answer,
> >
> > Regarding the main difference to Kessler et al, I am still a bit uncertain as to why the approach presented in this work performs significantly better.
> > Sure, modeling each weight separately gives more flexibility, but modeling whole rows or columns also seems reasonable, and the reduced complexity could generalize better. I think both approaches are valid and I would have not been surprised if both perform similar.
> >
> > So I am wondering if the performance difference could be due to other things, like annealing schedules etc.? Were the results of Kessler et al taken from their paper or did you compare it with your own implementation in which all other hyper-parameters are identical?

---

> > > ### Author Response · Authors · 2020-11-21
> > > **Thank you for your response**
> > >
> > > Thanks for your follow-up comments. We believe the improved performance as compared to Kessler et al could be due to the following additional reasons as well:
> > >
> > > 1. Masked Priors (section 3.4) provide our approach an additional flexibility by not having a Gaussian prior over weights that are never used.
> > > 2. As you mentioned, we use temperature annealing but their temperature parameter is fixed (roughly in the range of 0.7-1.0 range) for their experiments.
> > >
> > > We used our own implementation with identical hyper-parameters.
> > > Also, regarding the above differences, a more deep analysis with multiple configurations might be required to find out if the performance differences are due to combined effect or due to individual contribution of these modifications. Although this wasn’t the main focus of the paper, we can add an additional ablation study experiment for this in the final version. That said, please note that discriminative continual learning is only one of the goals that we share with Kessler et al (that too, with the above differences). They do not consider the generative (unsupervised) setting, or other aspects, such as task id inference. We would also like to point out that our work and Kessler et al have been independent and almost simultaneous developments (we can’t reveal more due to the requirements of double-blind reviewing).

---

> > > > ### Comment · AnonReviewer3 · 2020-11-23
> > > > **Thank you for your response**
> > > >
> > > > Thanks for your response,
> > > >
> > > > an ablation study regarding the masked prior would surely be insightful.
> > > > Please add this to the supplementary in the final version.
> > > >
> > > > I remain convinced that my score is appropriate and still suggest acceptance.

---

### Author Response · Authors · 2020-11-17
**Common Response to Reviewers**

We sincerely thank all the reviewers for the time taken in carefully assessing our work and for the overall positive feedback. The reviewers have generally found our Bayesian nonparametric approach for automated structure learning for continual learning to be theoretically sound, well motivated, and principled with strong experimental results.

A concern from AnonReviewer4 was regarding whether our method is for class-incremental or task-incremental setting. As the reviewer has guessed, it is indeed for the task-incremental setting. We apologize if this was not sufficiently clear in the original manuscript. In the uploaded revision, we have clarified this and also discussed in our response to AnonReviewer4 in detail. We hope our response will help clarify this and any of the other concerns that the reviewers may have.

Before addressing the individual comments from the reviewers, we provide a brief summary of the changes done in the revised version in order to incorporate the various comments from the reviewers.

##### Summary of changes in the new version.
------------
* As suggested by Reviewer 1 and Reviewer 2, we have corrected the typos and figure labels.
------------
Section 1
* As suggested by Reviewer 2, we have discussed the importance of structural learning; we will also update the title in the final version to reflect this.
------------
Section 3
* As suggested by Reviewer 4, we have made changes to highlight that we are doing task incremental learning.
------------
Section 4
* As suggested by Reviewer 3, we have added some discussion in the Related Work section on non-stationary data and the benefits of modelling network weights as opposed to activations using the IBP (as done in Kessler et al).
------------
Section 6
* Based on feedback from Reviewer 3 and Reviewer 4, we have updated our Conclusion section and mentioned class-incremental learning and non-stationary data as possible extensions for future work.
------------
Appendix
* As suggested by Reviewer 1, we have added analysis for model’s performance with varying network depth, and based on Reviewer 4’s suggestion, we have added a table for comparing model’s performance on a varying number of tasks with same architecture on permuted MNIST experiment.
* A discussion on why coresets are not helping much in performance for our model but does for VCL has been added in section H based on Reviewer 3’s feedback and we have also updated Table 2 to add comparisons with two more methods (single runs) as suggested in Reviewer 4’s comments.
* A comparison with HAT and VCL on cifar-10 and cifar-100 datasets has been added in section H.3 based on Reviewer 4's feedback.

---

### Decision · Program_Chairs · 2021-01-07
**Final Decision**

**Decision:**

Reject

**Comment:**

This paper proposes a Bayesian non-parametric method for task-incremental continual learning. It is more general than previous work in that it considers the network structure as a random variable and works for both supervised and unsupervised settings. Experimental results show that the proposed method outperforms prior work in the proposed tasks.

Pros:
- It is well motivated.
- It's theoretically sound.
- It can do task inference.
- It outperforms other methods in the proposed tasks.

Cons:
- The experimental setup was not very challenging, because the dataset(MNIST) was simple and the network was shallow.
- There was no ablation study to analyze the contributions of the algorithm to the performance.
- There is not enough experiments to support the advantage of task inference.
- The paper did not compare with the SOTA task-incremental learning algorithms HAT and DEN.

The main concerns of reviewers are on the experimental section as listed in cons and the difference from previous work. The authors explained that their method has the advantages over previous work
that it can do task inference. R3 agreed with the advantage and suggested more experiments in this direction should be performed.
The authors conducted additional experiments suggested by the reviewers including comparison with HAT. They also uploaded a revised version to incorporate the comments from the reviewers.

I think the paper is well motivated and the idea of applying Bayesian non-parametric for continuous learning is interesting. It could potentially motivate interesting future work on CL. However, the main advantages/contributions are not well presented and supported by the experiments. So at present time I believe there is much room for the authors to improve their paper before publication. I hope that the authors will be able to address the feedback they received to make this submission get where it should be.